# Islet autoantibodies as precision diagnostic tools to characterize heterogeneity in type 1 diabetes: a systematic review

Jamie L. Felton [1,2], Maria J. Redondo[3,4], Richard A. Oram[5,6,7], Cate Speake [8], S. Alice Long [9], Suna Onengut-Gumuscu [10], Stephen S. Rich [10], Gabriela S. F. Monaco[1,2], Arianna Harris-Kawano[1], Dianna Perez [1], Zeb Saeed[11], Benjamin Hoag[12], Rashmi Jain[12], Carmella Evans-Molina[1,2,11,13], Linda A. DiMeglio [1,2], Heba M. Ismail[1,2], Dana Dabelea[14], Randi K. Johnson[15,16], Marzhan Urazbayeva[3], John M. Wentworth [17,18,19], Kurt J. Griffin[12,20,200] & Emily K. Sims [1,2,200] ✉ On behalf of the ADA/EASD PMDI*

## Abstract

**Background** Islet autoantibodies form the foundation for type 1 diabetes (T1D) diagnosis and staging, but heterogeneity exists in T1D development and presentation. We hypothesized that autoantibodies can identify heterogeneity before, at, and after T1D diagnosis, and in response to disease-modifying therapies.

**Methods** We systematically reviewed PubMed and EMBASE databases (6/14/2022) assessing 10 years of original research examining relationships between autoantibodies and heterogeneity before, at, after diagnosis, and in response to disease-modifying therapies in individuals at-risk or within 1 year of T1D diagnosis. A critical appraisal checklist tool for cohort studies was modified and used for risk of bias assessment.

**Results** Here we show that 152 studies that met extraction criteria most commonly characterized heterogeneity before diagnosis (91/152). Autoantibody type/target was most frequently examined, followed by autoantibody number. Recurring themes included correlations of autoantibody number, type, and titers with progression, differing phenotypes based on order of autoantibody seroconversion, and interactions with age and genetics. Only 44% specifically described autoantibody assay standardization program participation.

**Conclusions** Current evidence most strongly supports the application of autoantibody features to more precisely define T1D before diagnosis. Our findings support continued use of pre-clinical staging paradigms based on autoantibody number and suggest that additional autoantibody features, particularly in relation to age and genetic risk, could offer more precise stratification. To improve reproducibility and applicability of autoantibody-based precision medicine in T1D, we propose a methods checklist for islet autoantibody-based manuscripts which includes use of precision medicine MeSH terms and participation in autoantibody standardization workshops.

## Plain language summary

Islet autoantibodies are markers found in the blood when insulin-producing cells in the pancreas become damaged and can be used to predict future development of type 1 diabetes. We evaluated published literature to determine whether characteristics of islet antibodies (type, levels, numbers) could improve prediction and help understand differences in how individuals with type 1 diabetes respond to treatments. We found existing evidence shows that islet autoantibody type and number are most useful to predict disease progression before diagnosis. In addition, the age when islet autoantibodies first appear strongly influences rate of progression. These findings provide important information for patients and care providers on how islet autoantibodies can be used to understand future type 1 diabetes development and to identify individuals who have the potential to benefit from intervention or prevention therapy.

---

A full list of affiliations appears at the end of the paper.   *A list of authors and their affiliations appears at the end of the paper.   ✉e-mail: eksims@iu.edu

Type 1 diabetes (T1D) results from the immune-mediated destruction of insulin-producing pancreatic beta cells[1]. Clinical disease is characterized by progressive hyperglycemia that, if left untreated, leads to ketoacidosis and death. T1D can be managed with exogenous insulin, and while technology surrounding glucose monitoring and insulin delivery have revolutionized diabetes care, effective disease management remains difficult, time-consuming, and costly. Islet autoantibodies that recognize insulin (IAA), glutamic acid decarboxylase (GADA), protein phosphatase-like IA-2 (IA-2A), zinc transporter 8 (ZnT8A), and islet cell cytoplasmic antigen (ICA), are well-validated predictors of risk and disease progression and have been proposed as diagnostic markers of presymptomatic stages of T1D. Stage 1 T1D is defined by the presence of multiple islet autoantibodies and normal glucose tolerance. This progresses to stage 2 T1D (multiple islet auto-antibodies and dysglycemia) and ultimately stage 3 T1D (meet American Diabetes Association (ADA) criteria for diabetes, usually with onset of clinical symptoms, typically requiring treatment with exogenous insulin)[2]. Understanding the pathophysiology that drives T1D progression through these stages remains critical to developing interventions to pause or reverse disease progression. However, vast heterogeneity exists in T1D progression, presentation, and responses to interventions. These differences suggest that differences in clinical features or presentation of disease progression and response to treatment could reflect discreet pathophysiological mechanisms. Along these lines, if distinct etiologic mechanisms are responsible for different forms of disease, it may be that specific subsets of individuals with T1D will respond better to specific disease-modifying therapies with improved risk/benefit ratios. Therefore, precision approaches to diagnosis may be necessary to effect disease modification in T1D.

The *Precision Medicine in Diabetes Initiative* (PMDI) was established in 2018 by the ADA in partnership with the European Association for the Study of Diabetes (EASD). The ADA/EASD PMDI includes global thought leaders in precision diabetes medicine who are working to address the burgeoning need for better diabetes prevention and care through precision medicine[3]. This T1D Diagnostics-focused Systematic Review is written on behalf of the ADA/EASD PMDI as part of a comprehensive evidence evaluation in support of the 2nd International Consensus Report on Precision Diabetes Medicine[4].

Multiple observational studies and clinical trials have investigated the impact of genetics, immune markers, metabolic function, and environmental factors, on the development and progression of T1D[5]. Some of these works have identified subgroups of individuals who may theoretically derive greater benefit than others from particular therapies. In this systematic review, we sought to identify aspects of precision medicine that have the potential to be adopted into clinical practice over the next 10 years. Given the substantial body of work focused on optimization, reproducibility, and validation of islet autoantibodies as biomarkers of islet autoimmunity[6–10], and their increased use in clinical practice since the development of the T1D staging system[2], we chose to focus on islet autoantibodies as an individual feature of disease. We explored and summarized evidence that islet auto-antibodies can be used to identify unique phenotypes of disease presentation and progression at four clinically-relevant timepoints: prior to clinical (stage 3) T1D diagnosis, at stage 3 T1D onset, after stage 3 T1D diagnosis, and in response to disease-modifying therapy at diagnosis (new onset trials) or before the time of stage 3 T1D diagnosis (prevention trials).

Here we show that autoantibody type/target was most frequently examined, followed by autoantibody number. Recurring themes included correlations of autoantibody number, type, and titers with progression, differing phenotypes based on order of autoantibody seroconversion, and interactions with age and genetics. Our findings suggest that the application of autoantibody features, specifically in relation to age and genetic risk, prior to diagnosis has the most potential to more precisely define and understand differences in T1D progression. To improve reproducibility and applicability of autoantibody-based precision medicine in T1D, we propose a methods checklist for islet autoantibody-based manuscripts which includes use of precision medicine MeSH terms and participation in autoantibody standardization workshops.

## Methods

### Data Source

We developed a search strategy using an iterative process that involved identification of Medical Subject Headings (MeSH) and text words, followed by refinement based on a sensitivity check for key articles identified by group members. On 10/25/21 "Precision Medicine"[Mesh] AND (Latent Auto-immune Diabetes in Adults [Mesh] OR "Diabetes Mellitus, Type 1"[Mesh]) was applied as an initial search strategy to PubMed. Based on identification of only 128 papers (mostly narrative reviews), the search strategy was expanded to include additional terms linked to precision medicine (Supplementary Note 1). This strategy was applied to PubMed and EMBASE databases on 6/14/2022.

### Study selection

The Covidence platform was used for stages of systematic review. To be included, studies must have involved individuals with high genetic risk (based on family history or genotype), single islet autoantibody positivity, stage 1 T1D (multiple islet autoantibody positivity and normal glucose tolerance), stage 2 T1D (positive islet autoantibodies and abnormal glucose tolerance), or stage 3 T1D (overt hyperglycemia, clinical symptoms of untreated T1D). Individuals with stage 3 T1D must have been within 1 year of diagnosis. Eligible study types included randomized controlled trials; systematic reviews or meta-analyses of randomized controlled trials; cross-sectional studies; open-label extension studies; prospective observational studies; retrospective observational studies; and post hoc analyses. Studies must have had a total sample size ≥10 per experimental or control group studied and have been published as a full paper in English in a peer-reviewed journal within 10 years of the search (2011–2022). Studies of non-T1D populations, unclearly classified diabetes populations, or mixed populations that included T1D among other diabetes types (type 2 diabetes, Latent Autoimmune Diabetes in Adults (LADA), gestational diabetes, or hypothetical cohort) were excluded. Several key articles identified by the group of experts that also met inclusion criteria but were not included in the search results because of search restrictions made to improve search feasibility, were also included in the analysis. In addition, clinical trials testing auto-antibody features associated with response to disease-modifying therapies from the last 25 years were also added, given the modest number of clinical trials identified during the specified search period. These papers are denoted in their respective tables and reference lists by an asterisk (*).

Investigators independently screened and reviewed each potentially relevant article according to preliminary eligibility criteria determined by members of the review team. For Level 1 screening two investigators per article screened each title and abstract. Discordant assessments were discussed and resolved by consensus or arbitration after consultation with a member of the review leadership team (JLF, RO, KJG, MJR, or EKS). In January of 2022, to improve review feasibility, the decision was made to limit articles at the Level 2 screening step using additional inclusion criteria. Here articles were further limited to exposures testing detection of abnormal islet autoantibodies (i.e., presence, total number, type, or titer) and addressing outcomes related to progression to multiple antibody positivity or diabetes, heterogenous presentation of disease, progression of C-peptide loss after diabetes develops, or response to treatment. For Level 2 screening of eligible articles, full texts were retrieved and reviewed by two independent reviewers using the inclusion/exclusion criteria. Discordant assessments were similarly discussed and resolved.

### Data extraction

Two independent investigators from the writing group extracted data from each article meeting inclusion criteria, with consensus determined by a member of the leadership team. Extracted data included details on participant characteristics, intervention outcomes, methods, and conclusions of precision analyses on disease progression or treatment response. Investigators performed quality assessments using a modified version of the Joanna Briggs Institute's critical appraisal checklist tool for cohort studies (https://jbi.global/critical-appraisal-tools) in tandem for each eligible study

**Table 1 | Summary of key themes**

| |
|---|
| Themes extracted from review of papers prior to T1D diagnosis |
| Risk for T1D progression increases with autoantibody number |
| Younger age at seroconversion results in faster progression |
| Islet autoantibody type (IAA, GADA, IA-2A, ZnT8, ICA) influences progression |
| The addition of islet autoantibodies improves performance of genetic risk prediction |
| Positive predictive value of autoantibody titer and affinity varies by autoantibody type |
| Specific autoantibody assay methods impact risk stratification |
| Themes extracted from review of papers at the time of T1D diagnosis |
| Type of autoantibody positive at diagnosis differs by age (children more often IAA positive, adults more often GAD positive) |
| Earlier seroconversion/diagnosis correlates to accelerated beta cell loss |
| Positivity for certain autoantibodies at diagnosis may be linked to specific genotypes or SNPs (GADA associated with HLA DR3; IAA associated with *INS* SNPs). |
| Higher numbers of positive autoantibodies more common in younger children |
| Themes extracted from review of papers following T1D diagnosis |
| Lower autoantibody titers and numbers are associated with greater residual C-peptide |
| In children, autoantibody type (IAA vs. GAD or IA-2) correlates with accelerated beta cell loss |
| Themes extracted from review of papers about response to treatment with disease-modifying therapies |
| Responses to treatments did not show consistent differences based on autoantibody type |
| Agents targeting a specific antigen in individuals who were positive for the corresponding specific antibody did not show reproducible efficacy across the primary populations tested |

to determine overall risk of bias. Discordant extractions and quality assessments were resolved as above.

### Data analysis and synthesis
Because of heterogeneity of included studies (i.e., design, population, exposure, and outcomes tested) we were unable to perform a meta-analysis. Instead, we provide a list of key themes extracted from all studies in Table 1, and completed summaries of relevant studies (Tables 2 and 3 and Supplementary Data 1 and 2).

The protocol of this review was registered at PROSPERO (CRD42022340047) prior to implementation (available at https://www.crd.york.ac.uk/prospero/display_record.php?ID=CRD42022340047).

### Reporting summary
Further information on research design is available in the Nature Portfolio Reporting Summary linked to this article.

## Results
### Literature search and screening results
Of the 11,192 papers evaluated, 152 ultimately met inclusion criteria for extraction (Fig. 1). We categorized studies based on clinically relevant timepoints assessed: 91 characterized differences in rates of progression and clinical features prior to stage 3 T1D and were categorized as "prior to diagnosis" (Supplementary Data 1); 44 assessed differences in metabolic or immune features at stage 3 T1D onset and were categorized as "at diagnosis" (Supplementary Data 2; abbreviations and references for supplementary data provided in Supplementary Data 4 and 5, respectively); 11 characterized metabolic decline after diagnosis and were categorized as "after diagnosis" (Table 2); and 13 assessed differences in responses to disease-modifying therapies tested in clinical trials and were categorized as "treatment response" (Table 3). Of note, some papers included multiple studies of several outcomes; therefore, a total of 159 studies were identified from 152 papers. While the size prohibited inclusion of the list of studies from the "prior to diagnosis" and "at diagnosis" periods in the main text, a list of key themes extracted from all studies is included as Table 1.

All papers in the analysis assessed established islet autoantibodies (IAA, GADA, IA-2A, ZnT8A, ICA) or novel autoantibodies targeting other islet autoantigens. Islet autoantibodies were a primary focus for a majority of

the papers identified (101/152, 66%), while others included autoantibody assessments as part of a larger precision analysis or clinical trial follow-up. The most frequent autoantibody feature studied was autoantibody type/target protein (137/152, 90%), followed by autoantibody number (98/152, 64%), autoantibody titer (50/152, 33%), age at seroconversion (40/152, 26%), rate of seroconversion from single to multiple autoantibodies (32/152, 21%), order of autoantibody appearance after seroconversion (28/152, 18%), novel islet autoantibody/epitope identification (13/152, 9%), and autoantibody affinity (6/152, 4%). Four of 152 papers (3%) assessed the use of different autoantibody assays to improve specificity of autoantibody testing.

Only 10/152 (7%) studies focused on a population that did not feature primarily European ancestry (described in Supplementary Table 1). In 110 studies, race and ethnicity were not reported, and those that did report race and ethnicity used inconsistent approaches to reporting (e.g., combined vs. separated race and ethnicity categories). Of studies that reported race and ethnicity, the median percentage of participants identifying as non-Hispanic white was 89% (IQR 84%–97%).

### Prior to diagnosis
The majority of the literature using autoantibody features to define heterogeneity in T1D focused on the period leading up to stage 3 T1D diagnosis (91/152, 60%). Studies included in the "prior to diagnosis" group are summarized in Supplementary Data 1, and key themes identified after systematic review of these papers are listed in Table 1. Of these, 85 evaluated longitudinal or cross-sectional cohorts (summarized in Table 4), typically testing differences in rates of diabetes progression. Median sample size was 510 (IQR 134–2239). Pediatric only populations were included in 61% (55/91); one study included only adults. The remainder (35/91, 38%) included pediatric and adult populations combined. Impact of age on findings was tested in 75/91 studies; with a significant impact of age reported in 85% (64/75). Assessment of islet autoantibody features during progression to T1D highlighted phenotypes characterized by age and genetic risk that were more clearly delineated with the addition of autoantibody type. Key recurring themes are summarized below.

**Risk for progression to stage 3 T1D increases with autoantibody number.** In 2013, a combined analysis of large birth cohorts (DAISY,

## Table 2 | Autoantibody (Aab) features characterize progression after T1D diagnosis

| Study | n | Age group | Aab feature assessed | Age impact? | Aab methods | Findings |
|---|---|---|---|---|---|---|
| Hameed 2011[52] | 367 | Pediatric | • Timing of Aab development | No | • ELISA; RIA; IIA; RBA<br>• Standardization program not described | • Persistent Aab- status at and after diagnosis associated with preserved residual C-peptide.<br>• Aab- children testing negative for monogenic diabetes frequently exhibited diabetogenic HLA. |
| Nielsen 2011 (Hvidovre Study Group on Childhood Diabetes)[86] | 257 | Pediatric | • Aab type<br>• Novel Aab or epitope | Yes | • IIA; RIA; RBA<br>• Standardization program: DASP | • ZnT8R epitope Aab+ more frequent if >5 years at diagnosis while ZnT8W Aab+ was not age-related.<br>• No relationship between ZnT8A+, ICA+, IA-2A+ at 1 month post-diagnosis and residual C-peptide at 12 months post-diagnosis.<br>• IAA+ or GADA+ at 1 month post-diagnosis associated with lower 12-month stimulated C-peptide.<br>• GADA+ associated with 12-month HbA1c. |
| Andersen 2012 (Danish Remission Phase Study)[87] | 129 | Pediatric | • Aab type<br>• Aab titer<br>• Novel Aab or epitope | Yes | RBA; IIA<br>• Standardization program: DASP | • All ZnT8A variant titers decreased over 12 months post-diagnosis.<br>• Higher arginine variant of ZnT8A associated with higher C-peptide over 12 months post-diagnosis.<br>• Positive correlation between all three ZnT8A variants and IA-2A titers over 12 months post-diagnosis (but not GADA or IAA). |
| Sorensen 2012 (Danish Registry for Childhood Diabetes)[88] | 260 | Pediatric | • Aab number<br>• Aab titer<br>• Aab type | Yes | • Methods not described<br>• Standardization program: DASP | • Reductions in IA-2A, ZnT8W, or ZnT8Q (but not ZnT8R or GADA) titers over 3-6 years post-diagnosis associated with higher likelihood of detectable C-peptide. |
| Chao 2013[89] | 247 | Pediatric and adult | • Aab type | No | • Radioligand assay<br>• Standardization program: DASP | • GADA+ more common than IA-2A+ at diagnosis in Chinese patients with acute onset T1D (56.3% vs. 32.8%).<br>• Most patients remained GADA+ or IA-2A+ during follow-up.<br>• C-peptide values higher in GADA- or IA-2A- patients vs. GADA+ or IA-2A+ at diagnosis, independently of whether Aab positivity persisted over time or not. |
| Ludvigsson 2013 (BDD)[55] | 4017 | Pediatric | • Aab type | Yes | • RIA<br>• Standardization program not described | • IAA+ associated with more rapid post-diagnosis C-peptide loss when controlling for age.<br>• No relation to GADA+ or IA-2A+ detected. |
| Pecheur 2014[90] | 242 | Pediatric | • Aab number<br>• Aab type | Yes | • Methods and standardization program not described | • Partial remission duration longer in single Aab+ vs. those with both GADA+ and IA-2A+. |
| Stoupa 2016[53] | 452 | Pediatric | • Aab type | Yes | • IIA; RBA<br>• Standardization program not described | • Mean C-peptide at 2 years post-diagnosis correlated with ICA- or IAA- at diagnosis in European ethnic groups (European Caucasian, Moghrabin Caucasian, Black African, and Mixed Origin). |
| Marino 2017[54] | 204 | Pediatric | • Aab number | Yes | • ELISA; RBA; IIA<br>• Standardization program not described | • Higher Aab+ number at diagnosis associated with lower rates of partial remission. |
| Camilo 2020[91] | 51 | Pediatric and adult | • Aab number<br>• Aab type | Yes | • Enzyme immunoassay<br>• Standardization program not described | • No significant difference in partial remission rates in GADA+ compared to IA-2A+ Brazilian children.<br>• HLA DRB1*0301-DQB1*0201 associated with lower IA-2A+ and higher remission rates. |
| Steck 2021 (TEDDY)[92] | 113 | Pediatric | • Aab number<br>• Aab type | Yes | • RBA<br>• Standardization program not described | • Higher Aab+ number at diagnosis associated with higher rate of C-peptide loss in univariate analysis.<br>• IA-2A+ or ZnT8A+ at diagnosis associated with higher rate of C-peptide loss in univariate analysis.<br>• Relationships no longer statistically significant in multivariate analysis including age, sex, and weight z-score. |

Methods and standardization program were listed as not described for studies either making no mention of methods or program participation or for studies that included a reference in the "Methods" section but did not specifically list the method or specifically mention standardization program participation.

*Aab* autoantibody, *BDD* better diabetes diagnosis, *FDR* first degree relative, *GAD* glutamic acid decarboxylase antibody, *IA-2* islet antigen-2 antibody, *IAA* insulin autoantibody, *ICA* islet cell autoantibody, *TEDDY* The Environmental Determinants of Diabetes in the Young, *ZnT8* zinc transporter antibody *ELISA* enzyme linked immunosorbent assay, *IIA* indirect immunofluorescence assay, *RIA* radioimmunoassay, *RBA* radiobinding assay, *IASP* islet autoantibody standardization program, *DASP* diabetes autoantibody standardization program.

**Table 3 | Autoantibody (Aab) features characterize heterogeneity in responses to disease-modifying therapy**

| Study | n | Age group | Population studied[a] | Aab feature assessed | Age impact? | Aab methods | Findings |
|---|---|---|---|---|---|---|---|
| Christie 2002[59]a | 97 | Pediatric and adult | • New onset | • Aab type | No | • RBA<br>• Combined autoantibody workshop | • Cyclosporin had no significant effect on frequency of IA-2A+.<br>• In IA-2A− participants Cyclosporin reduced insulin doses and increased C-peptide.<br>• IA-2A+, GADA− participants were most resistant to cyclosporin.<br>• No differential effects were observed for partitioning by ICA+ or IAA+. |
| Gale 2004 (ENDIT)[93]a | 552 | Pediatric and adult | • FDR (ICA+) | • Aab number<br>• Aab type | No | • Methods not described<br>• Combined autoantibody workshop | • No difference in time to T1D noted between nicotinamide and placebo groups when adjusted for Aab number.<br>• No evidence of a nicotinamide treatment effect in groups divided by Aab status. |
| Skyler 2005 (DPT-1)[94]a | 372 | Pediatric and adult | • FDR<br>• Single Aab+<br>• Multiple Aab+ | • Aab titer<br>• Aab type | Not available | • IIA; RIA<br>• Standardization program not described | • Oral insulin did not delay or prevent T1D progression in ICA+ and IAA+ relatives. |
| Näntö-Salonen 2008[95]a | 264 | Pediatric | • High genetic risk<br>• FDR<br>• Multiple Aab+ | • Aab number<br>• Aab titer<br>• Aab type | No | • IIA; RBA<br>• DASP | • Aab features did not impact ability of nasal insulin to delay or prevent T1D. |
| Pescovitz 2009 (TrialNet)[61]a | 87 | Pediatric and adult | • New onset | • Aab number<br>• Aab titer<br>• Aab type | No | • IIA; RBA<br>• Standardization program not described | • No significant differential treatment effect of rituximab among subgroups based on Aab+. |
| Wherrett 2011 (TrialNet)[57]a | 145 | Pediatric and adult | • New onset (GAD+) | • Aab type | No | • IIA; RBA<br>• Standardization program not described | • Subcutaneous GAD-alum did not preserve insulin secretion in GADA+ participants with recently diagnosed T1D. |
| Yu 2011 (TrialNet)[62] | 87 | Pediatric and adult | • New onset | • Aab number<br>• Aab titer<br>• Aab type | Not available | • RIA<br>• DASP | • Rituximab led to marked suppression of IAA for 1–3 years compared with placebo but had smaller effect on GADA, IA-2A, and ZnT8A.<br>• 40% of IAA+ individuals became IAA− with rituximab (vs. none with placebo).<br>• IAA levels lower for those who maintained C-peptide during 1st year after diagnosis independent of rituximab treatment. |
| Ludvigsson 2012 (Diamyd)[58]a | 334 | Pediatric and adult | • New onset | • Aab titer<br>• Aab type | No | • ELISA<br>• Standardization program not described | • In GADA+ new onset T1D, alum-formulated GAD65 treatment did not reduce loss of stimulated C-peptide vs. placebo.<br>• Stratification based on baseline GADA titer did not impact treatment response. |
| Herold 2013 (ITN-AbATE)[96] | 77 | Pediatric and adult | • New onset | • Aab type | No | • IIA; RIA<br>• Standardization program not described | • Significant reduction in ZnT8A titer (but not IA-2A, IAA, or GADA) in teplizumab-treated participants after 1 year but not after 2 years.<br>• Baseline individual Aab positivity did not predict response to teplizumab. |
| Aronson 2014 (DEFEND-1)[97] | 272 | Pediatric and adult | • New onset | • Aab number<br>• Aab type | No | • Methods and standardization program not described | • No impact of GADA+ or IA-2A+ or Aab number on Otelixizumab treatment effect. |
| Demeester 2015[98] | 80 | Pediatric and adult | • New onset | • Aab number<br>• Aab type | No | • RBA<br>• Immunology of Diabetes Workshop on Insulin Aabs | • Higher IAA levels associated with better preservation of beta cell function and lower insulin with anti-CD3 treatment.<br>• In multivariate analysis, IAA or the interaction of IAA and C-peptide independently predicted outcome together with treatment.<br>• During follow-up, anti-CD3 responders (i.e., IAA+ participants with preserved beta cell function) showed a less pronounced insulin-induced rise in IAA and lower insulin needs.<br>• GADA, IA-2A, and ZnT8A levels were not influenced by anti-CD3, and their changes showed no relationship with outcomes. |
| Krischer 2017 (TrialNet)[99] | 560 | Pediatric and adult | • FDR<br>• Second degree relative<br>• Multiple Aab+<br>• Other: third degree relative | • Aab number<br>• Aab titer<br>• Aab type | Not available | • IIA; RIA; microinsulin Aab assay<br>• Standardization program not described | • Among multiple Aab+ relatives with a high IAA titer, 7.5 mg/day oral insulin did not delay or prevent T1D development vs. placebo. |

**Table 3 (continued) | Autoantibody (Aab) features characterize heterogeneity in responses to disease-modifying therapy**

| Study | n | Age group | Population studied[a] | Aab feature assessed | Age impact? | Aab methods | Findings |
|---|---|---|---|---|---|---|---|
| Herold 2019 (TrialNet)[60] | 76 | Pediatric and adult | • Multiple Aab+ | • Aab type | No | • IIA; RBA<br>• Standardization program not described | • Response to teplizumab vs. placebo was greater if ZnT8A+, also if GADA+, or IAA+, and if IA-2A or IAA were negative. |

Methods and standardization program were listed as not described for studies either making no mention of methods or program participation or for studies that included a reference in the "Methods" section but did not specifically list the method or specifically mention standardization program participation.

*Aab* autoantibody, *AbATE* Autoimmunity-Blocking Antibody for Tolerance in Recently Diagnosed Type 1 Diabetes, *ENDIT* European Nicotinamide Diabetes Intervention Trial, *DEFEND-1* Durable Response Therapy Evaluation for Early or New-Onset Type 1 Diabetes, *DPT-1* Diabetes Prevention Trial Type 1, *FDR* first degree relative, *GAD* glutamic acid decarboxylase antibody, *IA-2* islet antigen-2 antibody, *IAA* insulin autoantibody, *ICA* islet cell autoantibody, *ITN* Immune Tolerance Network, *ZnT8* zinc transporter antibody, *ELISA* enzyme linked immunosorbent assay, *IIA* indirect immunofluorescence, *RIA* radioimmunoassay, *RBA* radiobinding assay, *IASP* islet autoantibody standardization program, *DASP* diabetes autoantibody standardization program.

[a]Participant groups were considered new onset if they were within 12 months of T1D diagnosis.

DIPP, BABYDIAB, BABYDIET) from different countries showed that T1D risk progressively increased with increasing numbers of positive autoantibodies. Of the 585 children who developed at least 2 autoantibodies, 84% developed type 1 diabetes within 15 years of follow-up[11]. This appreciation that lifetime risk of diabetes progression nears 100% once multiple positive islet autoantibodies have developed informs the current T1D staging system[2], and the impact of autoantibody number on risk of progression to stage 3 has been corroborated in numerous studies of additional cohorts[12–20].

**Younger age at seroconversion results in faster progression to stage 3 T1D.** Longitudinal assessment of over 2500 children with genetic risk for T1D followed in the DAISY cohort revealed that speed of progression to T1D diagnosis is strongly correlated with age at seroconversion[21]. These findings have been replicated in many subsequent T1D screening studies including the 2013 combined analysis of over 13,000 children from multiple birth cohorts mentioned above[11]. The ongoing TrialNet Pathway to Prevention natural history study has followed over 30,000 first and second degree relatives of individuals with T1D and shown that frequency of seroconversion from single to multiple autoantibody positivity decreases with age (cumulative incidence 2% for age 10 and under, 0.7% for those over 10 years)[22]. The clear relationship between younger age and faster progression was particularly strong prior to puberty[23]. A recent analysis from TEDDY described an exponential decline in risk and rate of development of single and multiple autoantibodies with increasing age[24].

**Islet autoantibody type (IAA, GADA, IA-2A, ZnT8A, ICA) influences progression.** In addition to autoantibody number, autoantibody type can be used to stratify risk more precisely for T1D progression. Overall, IA-2A and ZnT8A positivity have both been associated with increased T1D pathogenicity. Multiple studies described an increased risk of progression from single to multiple autoantibody positivity or to stage 3 T1D associated with IA-2A positivity[15,18,19,25–33]. This was most clearly seen in pediatric populations, as IA-2A positivity was preceded by or accompanied by development of other islet autoantibodies in 98% of the IA-2A positive children followed in the BABYDIAB cohort[29]. However, when pediatric and adult populations were evaluated together, ZnT8A positivity was most commonly associated with development of other autoantibodies, and in single autoantibody positive subjects, if the single autoantibody was ZnT8A, risk of progression to T1D was higher compared to single positivity for IAA, GADA, or IA-2A[34]. Positivity for IAA and GADA was more often associated with decreased risk or slower progression to T1D. In analysis of pediatric and adult subjects, IAA or GADA positive first degree relatives progressed more slowly to T1D than double autoantibody positive subjects positive for IA-2A and ZnT8A[15]. Reversion from single autoantibody positivity to autoantibody negativity was frequent for GADA and IAA, but not IA-2A and ZnT8A[14]. Multivariate analysis of subjects <20 years old showed that IA-2A, IAA, ICA, and ZnT8A positivity, but not GADA, could all independently predict diabetes progression[34].

**Order of autoantibody development varies by age and impacts risk for progression.** Multiple longitudinal studies have shown that the first autoantibody to appear differs significantly depending on age of seroconversion. Analysis from the Finnish Type 1 Diabetes Prediction and Prevention (DIPP) study showed that in children 2 years old and younger, abnormal IAA titers most frequently develop first, while children ages 3–5 years more frequently seroconvert to GADA positivity[35]. A smaller analysis from the Diabetes Auto Immunity Study in the Young (DAISY) cohort found that higher IAA levels were associated with younger age at diagnosis, and that nearly all young children who progressed to T1D were IAA positive[36]. Analysis of the BABYDIAB and BABYDIET pediatric cohorts also found that earliest autoantibody development (peak incidence 9 months) was most commonly

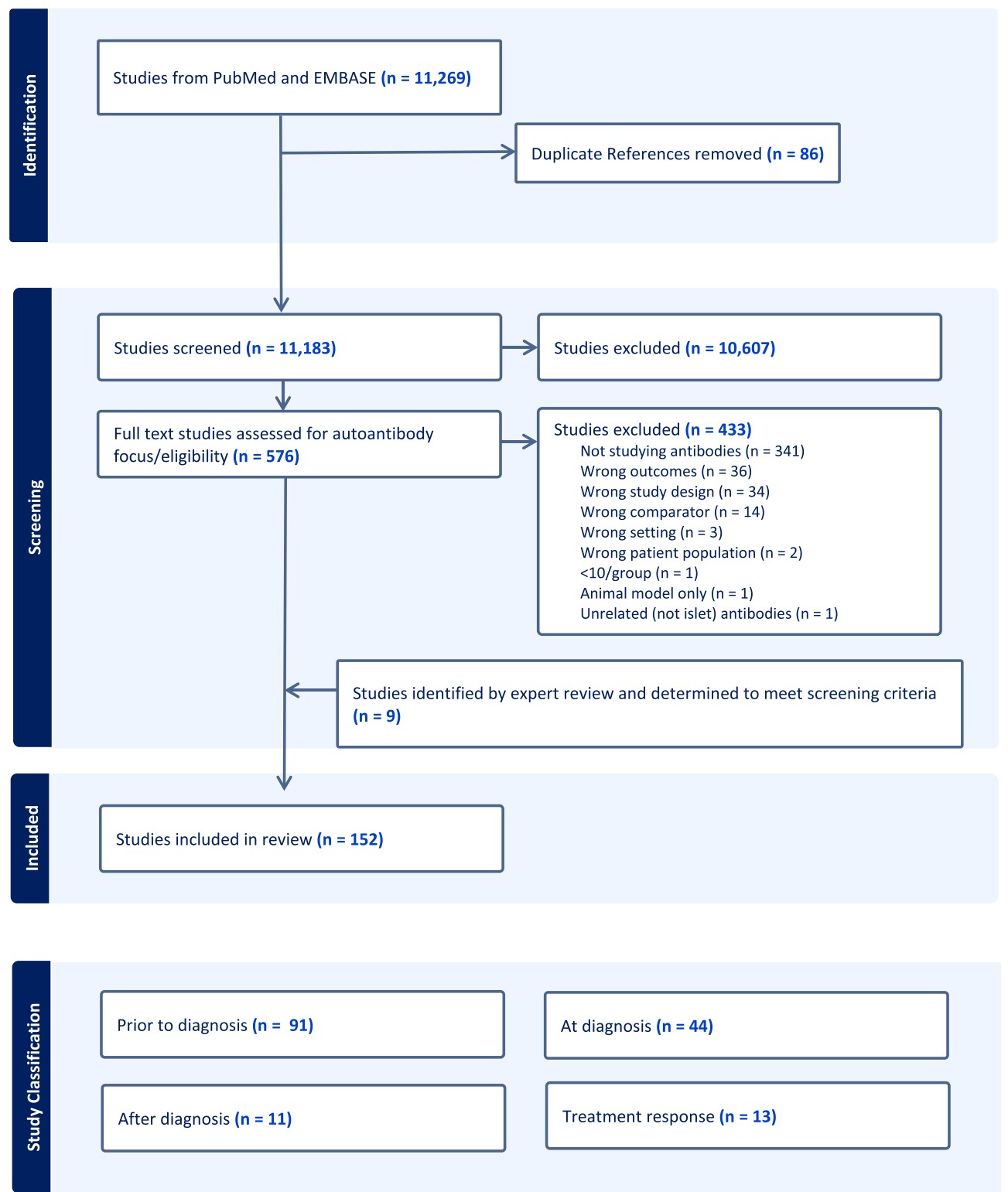

**Fig. 1 | PRISMA diagram.** For study classification, "Prior to diagnosis" refers to studies that assessed differences in rates of progression and clinical features during the period leading up to stage 3 T1D diagnosis. "At diagnosis" refers to studies that assessed heterogeneity in clinical features at the time of stage 3 T1D diagnosis. "After diagnosis" refers to studies that used features before or at the time of stage 3 T1D diagnosis to characterize subsequent metabolic decline (and preservation of endogenous insulin production). "Treatment response" refers to Studies that assessed heterogeneity in responses to disease-modifying therapies tested in clinical trials in subjects at or before stage 3 T1D diagnosis.

development of a single IAA which progressed to multiple autoantibodies. The Environmental Determinants of Diabetes in the Young (TEDDY) study has also suggested that IAA vs. GADA at seroconversion is linked to differing disease phenotypes, with additional features linked to IAA as the first autoantibody including specific single-nucleotide polymorphisms, male sex, father or siblings as a diabetic proband, introduction of probiotics at less than 1 month of age, and weight at 12 months[37]. Analysis of both pediatric and adult subjects combined

**Table 4 | Studies and cohorts referenced**

| Cohort/study name | Description |
|---|---|
| ABIS (All Babies in Southwest Sweden) | Prospective birth cohort study |
| BABYDIAB/BABYDIET | German prospective, longitudinal birth cohort |
| BDR (Belgian Diabetes Registry) | Registry of Belgian recent-onset diabetes patients and first-degree relatives recruited for longitudinal data and sample storage |
| DAISY (Diabetes Auto Immunity Study in the Young) | Prospective, longitudinal study |
| DEW-IT (Diabetes Evaluation in Washington Study) | Prospective, population-based observational study |
| DIPP (Diabetes Prediction and Prevention Study) | Prospective, population-based birth cohort |
| DiPiS (Diabetes Prediction in Skåne) | Prospective, longitudinal, population-based study |
| DPT-1 (Diabetes Prevention Trial Type 1) | Prospective, longitudinal study of relatives at risk for T1D |
| FPDR (Finish Pediatric Diabetes Register) | Cross-sectional registry of data and samples from individuals with new onset T1D and relatives |
| Fr1da (Early Detection for Early Care of Type 1 Diabetes) | Prospective cohort study |
| TEDDY (The Environmental Determinants of Diabetes in the Young) | Prospective birth cohort study |
| T1DI | Harmonized analysis of prospective cohort studies in Finland, Germany, Sweden, and the United States |
| TrialNet | TrialNet Pathway to Prevention natural history study/longitudinal cohort of first- and second-degree relatives at risk for T1D |

revealed that the risk of progression from single to multiple auto-antibodies decreased rapidly with increasing age when IAA was the first to develop. A decrease in risk with increasing age was also observed when GADA was the first to develop, but the risk reduction was less robust than that of IAA first[38].

**The addition of autoantibodies improves performance of genetic risk stratification to predict progression.** Highest genetic risk for T1D is associated with genes that encode MHC class II molecules[39]. Multiple studies suggest that genetic risk stratification with other identified risk variants has the potential to be improved by the consideration of autoantibody features. While MHC class II-associated genetic risk is well defined, less is known about risk associated with MHC class I genes. In a study of Belgian adult and pediatric first-degree relatives who were carriers of the high-risk MHC class II HLA-DQ2/DQ8, risk was further increased by the presence of MHC class I HLA-A*24 if subjects were also positive for IA-2A, but not if subjects were IA-2A negative. Additional screening for MHC class I HLA-B*18 with HLA-DQ2/DQ8, HLA-A*24, and IA-2A and/or ZnT8A increased the sensitivity of detecting rapid progressors[32]. For single autoantibody positive relatives, combinations of autoantibody positivity and high-risk alleles improved risk prediction, with younger age, HLA-DQ2/DQ8 genotype, and IAA positivity acting as independent predictors of more rapid seroconversion to multiple autoantibody positivity. The addition of autoantibody features to predict progression in genetically at-risk, multiple autoantibody positive relatives was less useful for this cohort, as most multiple autoantibody positive relatives progress to T1D within 20 years. Progression did occur more rapidly in the presence of IA-2A or ZnT8A, regardless of age, HLA-DQ genotype, and autoantibody number[15]. Among single and multiple autoantibody subjects, the non-HLA risk variant *PTPN22* risk allele (T/T) was associated with faster progression to T1D after appearance of the first and second autoantibodies, indicating a higher risk subgroup, while the *INS* risk allele had no impact on the risk of progression to T1D[40]. Performance of genetic risk scores calculated using multiple different genetic factors to predict disease progression were also shown to have the potential to be improved by the addition of autoantibody features. For example, the positive predictive value of a 30 T1D associated single-nucleotide polymorphism genetic risk score to predict T1D development in autoantibody positive individuals could be improved when the number of positive autoantibodies was also included in the model[41].

**Positive predictive value of autoantibody titer and affinity varies by type.** In addition to autoantibody type, autoantibody titers and affinities were also measured in many studies. However, only higher IAA and IA-2A titers and affinities have been shown to be linked to more rapid disease progression[42]. In pediatric and adult first degree relatives followed in the DPT-1 cohort, IA-2A titers increase and GADA titers decrease in the years prior to T1D diagnosis[28]. Similar findings were supported in young European children with HLA-DQB1-conferred disease susceptibility and advanced beta-cell autoimmunity where, in addition to young age, higher BMI SDS, and reduced first phase insulin response, higher IAA and IA-2A levels predicted T1D[43]. In persistently autoantibody positive children in the TEDDY study, higher mean IAA and IA-2A levels, but not GADA levels were associated with increased T1D risk[12,17]. The addition of islet autoantibody features to existing metabolic measures alone will likely be less impactful in stratifying risk, particularly after development of abnormalities in glucose tolerance. In the DPT-1 study cohort, the addition of autoantibody titers did not improve a prediction model based on oral glucose tolerance testing, and IAA titers did not provide significant prediction value in subgroups with abnormal glucose tolerance[44].

**Specific autoantibody assay methods impact risk stratification.** While autoantibody assay methods described were primarily radio-binding (RBA) or radioimmunoassays (RIA), other methods used included ELISA, other competitive and non-competitive binding assays, and indirect immunofluorescence. Four papers assessed differences between traditional radiobinding (RBA) assays and newer electrochemiluminescent (ECL) assays. Overall, this work suggested that ECL assays had higher positive predictive value, were more sensitive, and defined seroconversion earlier than traditional RBA assays[45–47]. This association was most pronounced in single autoantibody positive populations[46]. In the DAISY cohort, only 3 of 11 single autoantibody positive children testing positive for ZnT8A by RBA were also positive for ZnT8A by ECL. All 3 progressed to T1D, suggesting that ECL assays may identify a subset of higher risk, single autoantibody positive individuals[47].

**At diagnosis**
We identified 44 relevant studies that assessed the use of antibodies to define heterogeneous phenotypes at stage 3 T1D onset in the "at diagnosis" group (Supplementary Data 2). Median sample size was 561 (IQR 266–1036). Pediatric only populations were included in 28/44 (64%).

Multiple studies demonstrated differences in autoantibody type by age at diagnosis. Compared to children at onset, adults were less likely to be ICA positive or IA-2A positive; however, there were no differences in GADA positivity rates[48]. In Chinese individuals with T1D, children with acute onset T1D showed higher prevalence of IA-2A, ZnT8A, and multiple autoantibody positivity than adults, and children diagnosed under 10 years had the highest frequency of IA-2A positivity and multiple antibody positivity[49]. Studies from the DAISY cohort found that age at diagnosis is strongly correlated with age at seroconversion and IAA levels[21]; however, this does not hold true for adults, where GADA is more commonly positive at diagnosis[50]. Children who develop autoantibodies and progress to T1D early in life have less functional beta-cell mass and higher rates of diabetic ketoacidosis (DKA) at diagnosis. No specific positive autoantibody type (GADA, IA-2A, IAA, ZnT8A) was consistently associated with DKA severity in children[51].

### After diagnosis

A total of 11 studies that assessed the use of autoantibodies to characterize progression after diagnosis were identified and summarized in Table 2. In these studies, autoantibodies were identified at diagnosis, prior to initiation of insulin; therefore, IAA positivity reflects loss of self-tolerance, rather than immune response to exogenous insulin. All studies used C-peptide measures to assess endogenous insulin production. Median sample size was smaller for this group (247, IQR 129–367). The majority (9/11, 82%) included only pediatric populations.

Conclusions from this set of studies varied widely; however, a general theme was that less autoimmunity at diagnosis (as reflected by autoantibody titer and number) was more commonly associated with greater residual C-peptide and more pronounced partial remission. Persistent autoantibody negative status was associated with preserved residual C-peptide in multiple studies[52]. Mean C-peptide 2 years post-diagnosis was correlated with absence of ICA or IAA at diagnosis in European ethnic groups[53]. Higher antibody number at diagnosis was associated with lower rates of partial remission[54]. In a study of new onset pediatric patients, when controlling for age, IAA positivity was associated with more rapid C-peptide decline post-diagnosis, while no relation was identified for GADA or IA-2A positivity[55].

### Treatment response

The "response to treatment" group included 13 primary randomized controlled trials of disease-modifying therapies tested prior to or at stage 3 T1D diagnosis (Table 3). Three of these trials were designed to test agents targeting a specific antigen in individuals who were positive for specific autoantibodies, with overall negative findings. The TrialNet oral insulin study, designed to test a subgroup identified as part of the DPT-1 study, where individuals with high IAA titers exhibited significant delay in time to diabetes compared to placebo, ultimately did not show an impact of oral insulin on time to diabetes in this population overall[56]. Similarly, studies testing whether a GADA antigen-based immunotherapy was effective in GADA positive individuals did not identify a treatment response in the overall study population[57,58]. Responses to immunomodulatory therapies were frequently reported to potentially differ by autoantibody type. Cyclosporin immune suppression tended to work more poorly in IA-2A positive individuals but reduced insulin requirements and increased C-peptide secretion in IA-2A negative individuals[59]. In the teplizumab anti-CD3 prevention trial, treatment response was greater when ZnT8A was negative, while the presence or absence of other autoantibodies was not as strongly associated with clinical response[60]. The B cell-depleting agent rituximab suppressed IAAs compared with placebo but had a much smaller effect on all other antibodies[61,62]. However, analysis of whether IAA positivity was associated with treatment response was not done. Importantly, all these trials tested combined pediatric and adult populations without considering age effects in autoantibody subgroup analyses, although none identified a statistically significant impact of age itself on treatment efficacy.

### Risk of bias analyses

Reviewers performed assessments of specific metrics related to autoantibody assay quality as well as overall study design (Fig. 2). Metrics to assess performance of autoantibody assays are shown in Fig. 2a. Seventy-four percent (112/152) of studies applied the same assay to all participants tested; this either did not occur or was not clearly described in the remaining 26% (40/152) of studies. Methods used to measure autoantibodies were described in 72% (109/152) of papers. The 43 papers that did not give specific assay information typically either referenced another paper for methods (33/152, 22%) or did not focus on antibodies as a main outcome (8/152, 5%). About half (74/152, 49%) described characteristics of autoantibody assays utilized such as sensitivity, specificity, and assay variation. Papers that did not describe assay characteristics also most commonly referenced another paper for methods (43/74, 58%) or did not primarily focus on autoantibodies (20/74, 27%). Although references for antibody methods and standardization may have been included, only forty-four percent of total papers (67/152) specifically mentioned participation in an autoantibody standardization program in the manuscript text. Over half of the 85 papers that did not mention this type of program primarily focused on autoantibodies.

Quality assessments also touched on other aspects of study design. Reviewers judged that study participant groups were all recruited or identified from similar populations in most (134/152, 88%) papers. Confounding factors were presented in 87% (132/152) of papers, but only addressed in the analysis in 71% (108/152). Multiple analyses or comparisons were tested in the vast majority of papers (143/152, 94%), but only 24 of these (17%) described corrections for multiple comparisons. Statistical analyses were judged as clearly documented and able to be replicated in 85% (129/152) of papers, not clearly documented in 8% (12/152) and documented but with concerns raised for approach in 7% (11/152).

Diagnosis of T1D was considered an applicable outcome 91% (138/152) of papers and of these, 126/138 (91%) included valid and reliable measures of T1D diagnosis. For the 111 papers with dichotomous outcomes over a period of follow-up, 90% (100/111) clearly described methods to ensure that participants were free of the outcome at study start. 111 studies included longitudinal follow-up. Specific descriptions of the follow-up period were typically included; this was most commonly over >5 years (61/111, 55%), with 30% (33/111) followed for 2-5 years, and 11% (12/111) followed for <2 years. Duration of follow-up was clearly described in 106/111 (95%). Loss to follow up was much less commonly described. Specifics were only included in 29% (32/111) of applicable papers, and strategies to address loss to follow-up were only described in 19% (21/111) of applicable papers.

## Discussion

This work explored and summarized evidence that islet autoantibodies could be used to identify and define specific phenotypes prior to, at, and after stage 3 T1D diagnosis, and in response to disease-modifying therapy. We systematically reviewed the application of antibody measurements to define heterogeneity at diagnostic timepoints before and after the onset of clinically symptomatic disease in 152 papers published over the past 10 years. The large majority of studies identified assessed antibody features prior to diagnosis, suggesting that overall, the application of antibody features to T1D precision diagnostics will be most impactful on defining differences in T1D phenotypes during this period of disease development.

Although multiple individual features (immune signatures, genetics, metabolic measures) could be applied to differentiate disease phenotypes, in this effort, we chose to focus on autoantibodies because standardized measures are currently available and their implementation as precision diagnostic tools in T1D has the potential to be rapidly implemented. As a well-established marker of islet autoimmunity, autoantibodies benefit from prior harmonization efforts, existing standardization workshops that compare assays using clinical samples, and for the most-established assays, easy accessibility to clinicians[6–9]. Indeed, the application of autoantibody number as a precision diagnostic tool that stratifies future disease risk has moved beyond the T1D research field, as the with the T1D staging system[2]

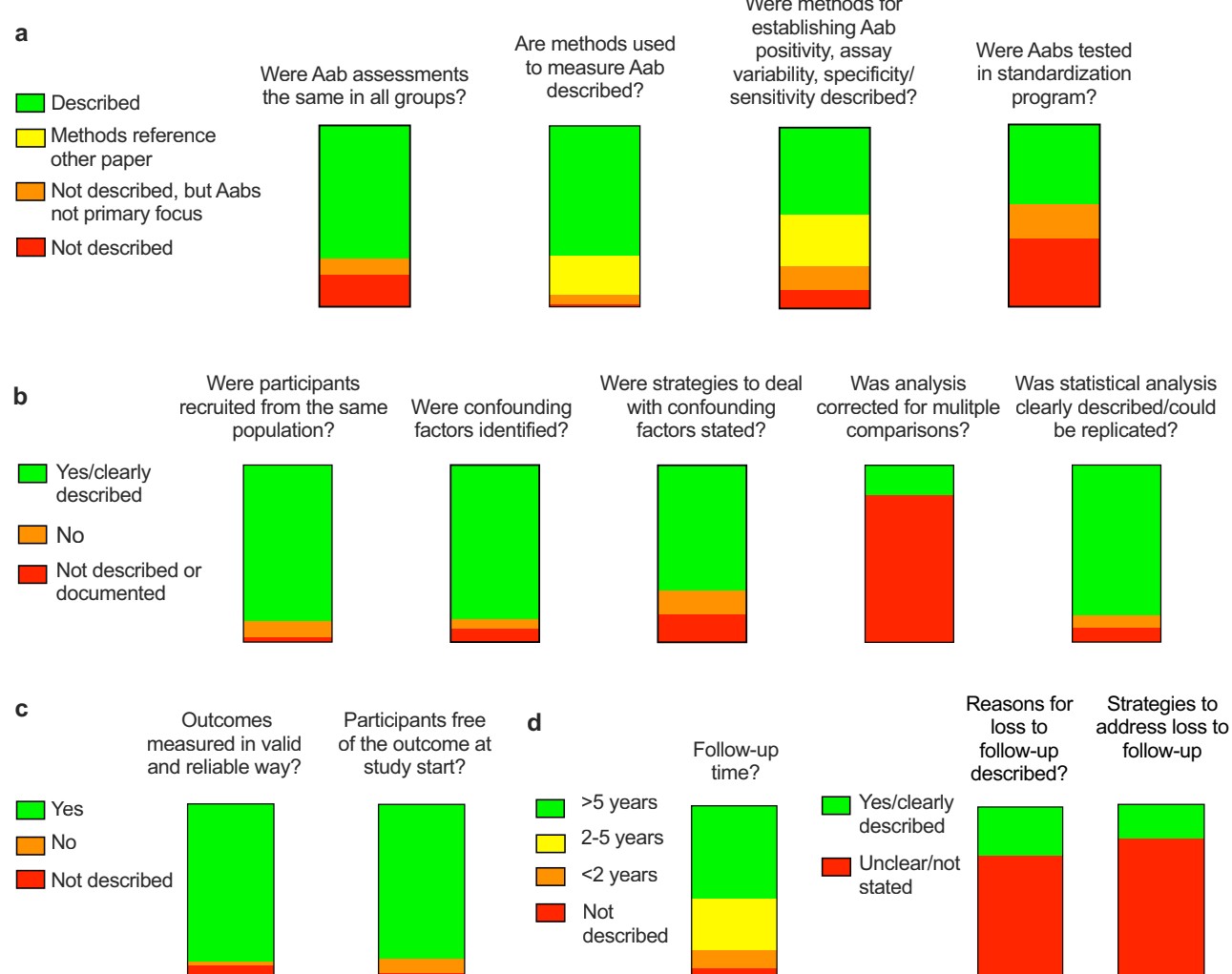

**Fig. 2 | Quality assessment of literature search results.** Graphical heat map of quality assessment questions surrounding (**a**) autoantibody measurements, (**b**) study design and analysis, (**c**) outcome assessments, and (**d**) study follow-up.

which is now being applied as part of clinical care guidelines to define stages of T1D[63]. Our review supports broad application of T1D staging using autoantibody number to guide individuals and clinicians on T1D risk. Furthermore, our findings support conclusions that have been drawn by others[64] that additional antibody features, such as antibody titer, type, and order of appearance, could be utilized together and in concert with auto-antibody number to more precisely stratify the current staging paradigm.

Our analysis supports existing evidence of the strong impact of age on heterogeneity of T1D development[65]. Specifically, our analysis confirms that younger age at seroconversion increases risk and rate for progression to stage 3 T1D, and this age-related risk can be further stratified using islet autoantibody type, titer, combined with number at seroconversion. The significant impact of age when considering use of autoantibodies to stratify risk suggests that (1) recommendations for use of autoantibodies in screening and prediction studies will need to consider stratification by age groups, and (2) that the analytic approach to autoantibody studies should include strategies to address impacts of age.

While most studies on autoantibodies in the period before Stage 3 T1D diagnosis focused on autoantibody number, type, or timing of ser-oconversion, fewer studies that passed our criteria for review assessed the immune responses that drive these changes. Therefore, there is continued need to understand how and when tolerance is broken in the context of clinical studies and how this leads to heterogeneous phenotypes. The few studies that did assess immune signatures[66–69] in multiple autoantibody

positive relatives revealed both proinflammatory and partially regulated (protective) phenotypes, which were also associated with autoantibody number. Interestingly, autoantibody negative relatives were characterized by the partially regulated phenotype[66], suggesting that progression to T1D may be the result of insufficient suppressive mechanisms, rather than differences in antigen targets. Immunoregulatory signatures were also identified in high HLA-risk siblings of subjects with T1D who were auto-antibody negative[70], though this study was excluded from our review due to sample size.

Precision diagnostics has particular utility in stratifying risk beyond autoantibody number in single autoantibody positive individuals, a group that is considered lower risk for T1D progression overall, and consequently, often do not meet inclusion criteria for clinical trials that require multiple positive autoantibodies. Studies of ECL vs. RBA assays suggest that ECL assays can identify a subset of higher risk, single autoantibody positive individuals. Autoantibody type was identified in this review as a common approach to stratify risk among single autoantibody positive individuals. For example, given the rarity of IA-2A as the initial autoantibody at ser-oconversion in birth cohorts, individuals who are cross-sectionally single autoantibody positive for IA-2A may reflect a higher risk group that has reverted to single autoantibody positivity, and are at higher risk of pro-gression to multiple autoantibody positivity and ultimately T1D. The T1DI analysis of over 24,000 children at increased genetic risk for T1D from prospective cohort studies in Finland, Germany, Sweden, and the US,

revealed that HLA-DR-DQ genotypes can stratify risk progression among children retaining a single autoantibody[71].

Evidence for use of antibodies at, after, and in response to disease-modifying therapies was less robust, and far fewer studies were identified. At diagnosis, the presence or absence of specific islet autoantibodies was also correlated with age, which might be expected given the differences in autoantibody presentation at seroconversion. However, multiple studies suggested that the primary autoantibodies at seroconversion had often disappeared at the time of diagnosis[14,72-75]; this is an open question in the field. While some evidence suggested that declining islet autoantibody titers and numbers after diagnosis corresponded to preserved residual C-peptide, we did not find convincing evidence to support the use of islet auto-antibodies to define heterogeneity in metabolic outcomes after stage 3 diagnosis. Evidence for use of islet autoantibody features to predict responses to disease-modifying therapies was modest. One potential explanation for this finding could be epitope spreading and neoantigen expansion that accompanies T1D disease progression, making the impact of a specific antigen (and its corresponding autoantibody) less significant by the time an individual has reached more advanced stages of disease.

Of note, our initial search strategy that targeted papers with combined use of MeSH (Medical Subject Headings) terms for "Precision Medicine" and "Type 1 Diabetes" identified only a small number of papers which were predominantly commentaries or reviews with very few original research articles. This likely represents the relatively recent application of precision medicine concepts to the field as well as a broader issue surrounding neb-ulous definitions of precision medicine[76]. Moving forward, inclusion of "Precision Medicine" as a MeSH term in manuscripts focused on T1D heterogeneity, stratification, or endotypes will be critical to allow researchers to easily access relevant studies in this area.

Many of the studies we reviewed emanated from prospective longitudinal cohort studies either from prevention trials or natural history cohorts, such as DPT-1, DAISY, BABYDIAB, TEDDY, Fr1da, DIPP, and the TrialNet Pathway to Prevention study. Likely related to this, overall, outcomes were judged to be reliably ascertained, and participants had substantial durations of follow-up (55% documented follow-up beyond 5 years). These qualities highlight the exceptional value that natural history cohorts have brought to the field of T1D precision medicine overall. However, our review has also highlighted quality concerns that will benefit from being addressed moving forward. An area that is particularly high yield is the use of autoantibody standardization workshops aimed at improving the performance and concordance of immunoassays used to measure islet autoantibodies. Despite standardi-zation programs being available throughout the timeframe studied in this review[77], participation in theses workshops was not uniformly described, even among papers with autoantibodies as a primary focus. Especially with more novel assays, clear reporting of methods and validation efforts are critical to the reproducibility of findings[6]. The fact that the framework has already been set to do this through the establishment of existing standardization programs makes more consistent and explicitly identi-fied participation in these workshops "low hanging fruit" for improve-ment of study quality.

Loss to follow-up in longitudinal studies was not frequently docu-mented. While analysis strategies frequently addressed differences in follow-up duration, systematic differences in loss to follow-up amongst different populations could theoretically still impact findings. Additionally, given the frequent reporting of interactions of autoantibody findings with age, con-sideration of relationships with age and other confounding factors is critical. This did not appear to be addressed in 30% of papers reviewed. Finally, many of the analyses of autoantibody subgroups in clinical trials were post hoc assessments, such that robust association of features with treatment response will require confirmation in a trial specifically designed for these subgroups.

The vast majority of studies assessed emanated from cohorts that were composed of groups of individuals of primarily European ancestry. Specific reporting on race and ethnicity were uncommon (only present in about ¼ of papers) and were inconsistently applied. Validation of antibodies as a tool for precision diagnosis across diverse populations, such as has been per-formed with genetic T1D risk scores[78] will be important to ensure broader applicability.

There are some limitations to this analysis. While our original goal was a meta-analysis of studies conducted at each time point, given the significant heterogeneity of exposures, outcomes, and study conditions, we were lim-ited to a systematic review of the state of the literature. Mainly for review feasibility, we limited our search to a 10-year period and to outcomes related to islet autoantibodies. Because of this, important papers in the field that did not meet inclusion criteria or published earlier or later than our search were not included as part of this review. Since the first estimation of T1D risk using ICA and HLA in relatives of individuals with T1D in the 1988 analysis of the Barts-Windsor Family study[79], leaders in the field of T1D prediction have been using autoantibody features to stratify T1D risk. This review indeed stands upon the shoulders of giants whose work was recently ele-gantly highlighted by Bonifacio and Achenbach in their 2019 review of islet autoantibody history[80]. Importantly, the key finding that the use of auto-antibodies for application of precision medicine is highest yield in the period prior to T1D diagnosis is supported by papers published prior to 2011.

More recently published papers were not included based on timing of our literature search, but are nonetheless important. For example, the Fr1da-study group recently published data that showed IA-2A positivity and titer, in combination with hemoglobin A1c and OGTT glucose values, could be used to generate a progression likelihood score that effectively identified presymptomatic multiple autoantibody positive children at very high risk of progression to clinical disease[81]. More recently, multiple studies generated from the harmonized data of five prospective cohorts into the combined T1DI cohort have been published, highlighting important insights gained from analysis of this cohort, including how the stringency of the definition of multiple Aab positivity markedly alters the risk of progression[82], suggesting that initial screening for islet autoantibodies at two ages (2 and 6 years) may be a sensitive and efficient approach to T1D population screening[83], and identifying trajectories of progression based on autoantibody positivity and titer[84,85]. We anticipate that data generated from this powerful, harmonized cohort will continue to have important implications on the application of autoantibody status to precision diagnostics.

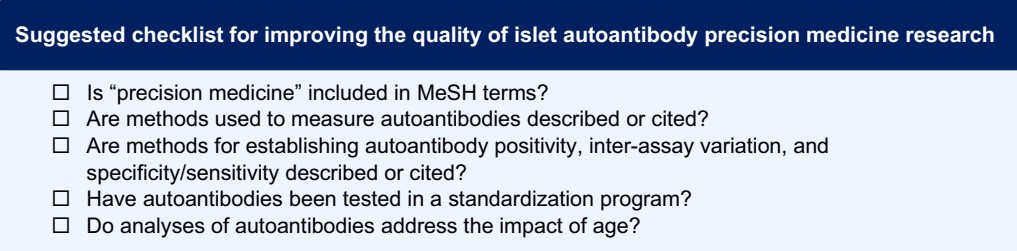

**Suggested checklist for improving the quality of islet autoantibody precision medicine research**

☐ Is "precision medicine" included in MeSH terms?
☐ Are methods used to measure autoantibodies described or cited?
☐ Are methods for establishing autoantibody positivity, inter-assay variation, and specificity/sensitivity described or cited?
☐ Have autoantibodies been tested in a standardization program?
☐ Do analyses of autoantibodies address the impact of age?

**Fig. 3 | Questions to consider to improve the quality of and applicability of islet autoantibody precision medicine research.** Based on findings of this review, this is a suggested checklist for improving the quality of islet autoantibody precision medicine research.

Our review inevitably excluded populations of adult individuals with T1D who have been historically misclassified as having type 2 diabetes. Along these lines, we decided not to include LADA in this review due to significant inconsistencies in the definition of LADA in many studies, but this group of individuals certainly contribute to T1D heterogeneity. Whether the findings noted here can be replicated in studies of LADA populations deserves future study/review.

Notwithstanding these limitations, overall, our findings suggest that islet autoantibodies are likely to be most useful to define T1D heterogeneity prior to clinical diagnosis, supporting prior efforts to use autoantibodies as part of precision T1D staging. Further benefit may be gained by their incorporation into risk scores that include features beyond autoantibody number and also consider age and genetics. Moving forward, thoughtfully designed, prospective analyses to test these relationships with disease-modifying therapies will be critical for further application of these observations and development of precision medicine approaches to T1D disease-modifying therapies. To aid in potential precision application and assessment of potential risk of bias, in Fig. 3 we provide a suggested checklist for studies applying islet autoantibodies to phenotypic heterogeneity before and after T1D diagnosis. Additionally, systematic review of other individual features, such as genetics, metabolic function, and other immune findings, will likely provide further insight into the current evidence for strategies to apply these features to precision T1D diagnostics.

## Data availability

All studies reviewed were identified and can be accessed via publicly available databases (PubMed and Embase). Source data can be found in Supplementary Data 3. A full list of included studies is available in Supplementary Data 6. Article review data supporting the findings of this study are available upon reasonable request from the corresponding author.

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

## Acknowledgements

We thank Krister Aronsson and Maria Bjorklund from Lund University for assistance with database searches and Russell de Souza from McMaster University for advice on critical appraisal. The ADA/EASD Precision Diabetes Medicine Initiative, within which this work was conducted, has received the following support: The Covidence license was funded by Lund University (Sweden) for which technical support was provided by Maria Björklund and Krister Aronsson (Faculty of Medicine Library, Lund University, Sweden) and Russell de Souza from McMaster University for advice on critical appraisal. Administrative support was provided by Lund University (Malmö, Sweden), University of Chicago (IL, USA), and the American Diabetes Association (Washington D.C., USA). The Novo Nordisk Foundation (Hellerup, Denmark) provided grant support for in-person writing group meetings (PI: L Phillipson, University of Chicago, IL). J.L.F.: DiabDocs K12 program 1K12DK133995-01 (DiMeglio, Maahs PIs), The Leona M. & Harry B. Helmsley Charitable Trust Grant #2307-06126 (Felton PI); M.J.R.: NIH NIDDK R01DK124395; R.A.O.: R.A.O. had a UK MRC confidence in concept award to develop a type 1 diabetes GRS biochip with Randox R&D and has ongoing research funding from Randox; and has research funding from a Diabetes UK Harry Keen Fellowship (16/0005529), National Institute of Diabetes and Digestive and Kidney Diseases grants (NIH R01 DK121843–01 and U01DK127382–01), JDRF (3-SRA-2019–827-S-B, 2-SRA-2022–1261-S-B, 2-SRA-2002–1259-S-B, 3-SRA-2022–1241-S-B, and 2-SRA-2022–1258-M-B), and The Larry M and Leona B Helmsley Charitable Trust; and is supported by the National Institute for Health and Care Research Exeter Biomedical Research Centre. The views expressed are those of the author(s) and not necessarily those of the National Institutes for Health Research or the Department of Health and Social Care; S.A.L.: NIH NIAID R01 AI141952 (PI), NIH NCI R01 CA231226 (Other support), NIH NIAID 1 R01HL149676 (Other support), NIH NIDDK 1UC4DK117483 (subaward), JDRF 3-SRA-2019-851-M-B; S.O.G.: NIH R01 DK121843–01; S.R.: R01 DK122586, The Leona M and Harry B Helmsley Charitable Trust 2204-05134; C.E.M.: R01DK093954, R01DK127236, U01DK127786, R01DK127308, and UC4DK104166, U54DK118638, P30 P30 DK097512), a US Department of Veterans Affairs Merit Award (I01BX001733), grants from the JDRF (3-IND-2022-1235-I-X) and Helmsley Charitable Trust (2207-05392), and gifts from the Sigma Beta Sorority, the Ball Brothers Foundation, and the George and Frances Ball Foundation;

L.A.D.: NIH for TrialNet U01DK106993/6163-1082-00-BO, DiabDocs K12 program 1K12DK133995-01, CTSI UL1TR001108-01; H.I.: K23DK129799; R.J.: NIH R03-DK127472 and The Leona M. and Harry B. Helmsley Charitable Trust (2103-05094); J.W.: JDRF 2-SRA-2022-1282-M-X, 3-SRA-2022-1095-M-B, 4-SRA-2022-1246-M-N, 3-SRA-2023-1374-M-N; K.G.: The Leona M. and Harry B. Helmsley Charitable Trust and Sanford Health. E.K.S.: R01DK121843; R01DK121929A1, R01DK133881, U01DK127786, U01 DK127382 Effort from this grant (to E.K.S., H.I., J.F.) is also supported by Grant 2021258 from the Doris Duke Charitable Foundation through the COVID-19 Fund to Retain Clinical Scientists collaborative grant program and was made possible through the support of Grant 62288 from the John Templeton Foundation.

## Author contributions

J.L.F., K.J.G., R.A.O., M.J.R. and E.K.S. designed the project, performed systematic review, interpreted results, and wrote and edited the manuscript. C.S., S.A.L., S.O.G., S.S.R., G.S.F.M., A.H.K., D.P., Z.S., B.H., R.J., C.E.M., L.A.D., H.M.I., D.D., R.K.J., M.U. and J.M.W. contributed to design of the project, performed systematic review, and edited the manuscript.

## Competing interests

E.K.S. has received compensation for educational lectures from Medscape, ADA, and MJH Life Sciences and as a consultant for DRI Healthcare. C.E.M. reported serving on advisory boards for Provention Bio, Isla Technologies, MaiCell Technologies, Avotres, DiogenyX, and Neurodon; receiving in-kind research support from Bristol Myers Squibb and Nimbus Pharmaceuticals; and receiving investigator initiated grants from Lilly Pharmaceuticals and Astellas Pharmaceuticals. L.A.D. reports research support to institution from Dompe, Lilly, Mannkind, Provention, Zealand and consulting relationships with Abata and Vertex. R.A.O. had a UK MRC Confidence in concept grant to develop a T1D GRS biochip with Randox Ltd, and has ongoing research funding from Randox R&D. No other authors report any relevant conflicts of interest.

## Additional information

[1]Department of Pediatrics, Center for Diabetes and Metabolic Diseases, Indianapolis, IN, USA. [2]Herman B Wells Center for Pediatric Research, Indiana University School of Medicine, Indianapolis, IN, USA. [3]Department of Pediatrics, Baylor College of Medicine, Houston, TX, USA. [4]Division of Pediatric Diabetes and Endocrinology, Texas Children's Hospital, Houston, TX, USA. [5]NIHR Exeter Biomedical Research Centre (BRC), Academic Kidney Unit, University of Exeter, Exeter, UK. [6]Department of Clinical and Biomedical Sciences, University of Exeter Medical School, Exeter, UK. [7]Royal Devon University Healthcare NHS Foundation Trust, Exeter, UK. [8]Center for Interventional Immunology, Benaroya Research Institute, Seattle, WA, USA. [9]Center for Translational Immunology, Benaroya Research Institute, Seattle, WA, USA. [10]Center for Public Health Genomics, University of Virginia, Charlottesville, VA, USA. [11]Department of Endocrinology, Diabetes and Metabolism, Indiana University School of Medicine, Indianapolis, IN, USA. [12]Department of Pediatrics, Sanford School of Medicine, University of South Dakota, Sioux Falls, SD, USA. [13]Richard L. Roudebush VAMC, Indianapolis, IN, USA. [14]Lifecourse Epidemiology of Adiposity and Diabetes (LEAD) Center, Aurora, CO, USA. [15]Department of Biomedical Informatics, University of Colorado Anschutz Medical Campus, Aurora, CO, USA. [16]Department of Epidemiology, Colorado School of Public Health, Aurora, CO, USA. [17]Royal Melbourne Hospital Department of Diabetes and Endocrinology, Parkville, VIC, Australia. [18]Walter and Eliza Hall Institute, Parkville, VIC, Australia. [19]University of Melbourne Department of Medicine, Parkville, VIC, Australia. [20]Sanford Research, Sioux Falls, SD, USA. [200]These authors contributed equally: Kurt J. Griffin, Emily K. Sims. *A list of authors and their affiliations appears at the end of the paper. ✉e-mail: eksims@iu.edu

## the ADA/EASD PMDI

Deirdre K. Tobias[21,22], Jordi Merino[23,24,25], Abrar Ahmad[26], Catherine Aiken[27,28], Jamie L. Benham[29], Dhanasekaran Bodhini[30], Amy L. Clark[31], Kevin Colclough[6], Rosa Corcoy[32,33,34], Sara J. Cromer[24,35,36], Daisy Duan[37], Jamie L. Felton[2,38,39], Ellen C. Francis[40], Pieter Gillard[41], Véronique Gingras[42,43], Romy Gaillard[44], Eram Haider[45], Alice Hughes[6], Jennifer M. Ikle[46,47], Laura M. Jacobsen[48], Anna R. Kahkoska[49], Jarno L. T. Kettunen[50,51,52], Raymond J. Kreienkamp[24,25,35,53], Lee-Ling Lim[54,55,56], Jonna M. E. Männistö[57,58], Robert Massey[45], Niamh-Maire Mclennan[59], Rachel G. Miller[60], Mario Luca Morieri[61,62], Jasper Most[63], Rochelle N. Naylor[64], Bige Ozkan[65,66], Kashyap Amratlal Patel[6], Scott J. Pilla[67,68], Katsiaryna Prystupa[69,70], Sridharan Raghavan[71,72], Mary R. Rooney[65,73], Martin Schön[69,70,74], Zhila Semnani-Azad[22], Magdalena Sevilla-Gonzalez[35,36,75], Pernille Svalastoga[76,77], Wubet Worku Takele[78], Claudia Ha-ting Tam[56,79,80], Anne Cathrine B. Thuesen[23], Mustafa Tosur[3,4,81], Amelia S. Wallace[65,73], Caroline C. Wang[73], Jessie J. Wong[82], Jennifer M. Yamamoto[83], Katherine Young[6], Chloé Amouyal[84,85], Mette K. Andersen[23], Maxine P. Bonham[86], Mingling Chen[87], Feifei Cheng[88], Tinashe Chikowore[36,89,90,91], Sian C. Chivers[92], Christoffer Clemmensen[23], Dana Dabelea[93], Adem Y. Dawed[45], Aaron J. Deutsch[25,35,36], Laura T. Dickens[94], Linda A. DiMeglio[2,38,39,95], Monika Dudenhöffer-Pfeifer[26],

Carmella Evans-Molina[2,13,38,39], María Mercè Fernández-Balsells[96,97], Hugo Fitipaldi[26], Stephanie L. Fitzpatrick[98], Stephen E. Gitelman[99], Mark O. Goodarzi[100,101], Jessica A. Grieger[102,103], Marta Guasch-Ferré[22,104], Nahal Habibi[102,103], Torben Hansen[23], Chuiguo Huang[56,79], Arianna Harris-Kawano[2,38,39], Heba M. Ismail[2,38,39], Benjamin Hoag[105,106], Randi K. Johnson[15,16], Angus G. Jones[6,7], Robert W. Koivula[107], Aaron Leong[24,36,108], Gloria K. W. Leung[86], Ingrid M. Libman[109], Kai Liu[102], S. Alice Long[9], William L. Lowe Jr.[110], Robert W. Morton[111,112,113], Ayesha A. Motala[114], Suna Onengut-Gumuscu[115], James S. Pankow[116], Maleesa Pathirana[102,103], Sofia Pazmino[117], Dianna Perez[2,38,39], John R. Petrie[118], Camille E. Powe[24,35,36,119], Alejandra Quinteros[102], Rashmi Jain[12,120], Debashree Ray[73,121], Mathias Ried-Larsen[122,123], Zeb Saeed[124], Vanessa Santhakumar[21], Sarah Kanbour[67,125], Sudipa Sarkar[67], Gabriela S. F. Monaco[2,38,39], Denise M. Scholtens[126], Elizabeth Selvin[65,73], Wayne Huey-Herng Sheu[127,128,129], Cate Speake[8], Maggie A. Stanislawski[15], Nele Steenackers[117], Andrea K. Steck[130], Norbert Stefan[70,131,132], Julie Støy[133], Rachael Taylor[134], Sok Cin Tye[135,136], Gebresilasea Gendisha Ukke[78], Marzhan Urazbayeva[4,137], Bart Van der Schueren[117,138], Camille Vatier[139,140], John M. Wentworth[17,18,19], Wesley Hannah[141,142], Sara L. White[92,143], Gechang Yu[56,79], Yingchai Zhang[56,79], Shao J. Zhou[103,144], Jacques Beltrand[145,146], Michel Polak[145,146], Ingvild Aukrust[76,147], Elisa de Franco[6], Sarah E. Flanagan[6], Kristin A. Maloney[148], Andrew McGovern[6], Janne Molnes[76,147], Mariam Nakabuye[23], Pål Rasmus Njølstad[76,77], Hugo Pomares-Millan[26,149], Michele Provenzano[150], Cécile Saint-Martin[151], Cuilin Zhang[152,153], Yeyi Zhu[154,155], Sungyoung Auh[156], Russell de Souza[112,157], Andrea J. Fawcett[158,159], Chandra Gruber[160], Eskedar Getie Mekonnen[161,162], Emily Mixter[163], Diana Sherifali[112,164], Robert H. Eckel[165], John J. Nolan[166,167], Louis H. Philipson[163], Rebecca J. Brown[156], Liana K. Billings[168,169], Kristen Boyle[93], Tina Costacou[60], John M. Dennis[6], Jose C. Florez[24,25,35,36], Anna L. Gloyn[46,47,170], Maria F. Gomez[26,171], Peter A. Gottlieb[130], Siri Atma W. Greeley[172], Kurt Griffin[12,20], Andrew T. Hattersley[6,7], Irl B. Hirsch[173], Marie-France Hivert[24,174,175], Korey K. Hood[82], Jami L. Josefson[158], Soo Heon Kwak[176], Lori M. Laffel[177], Siew S. Lim[78], Ruth J. F. Loos[23,178], Ronald C. W. Ma[56,79,80], Chantal Mathieu[41], Nestoras Mathioudakis[67], James B. Meigs[36,108,179], Shivani Misra[180,181], Viswanathan Mohan[182], Rinki Murphy[183,184,185], Richard Oram[6,7], Katharine R. Owen[107,186], Susan E. Ozanne[187], Ewan R. Pearson[45], Wei Perng[93], Toni I. Pollin[148,188], Rodica Pop-Busui[189], Richard E. Pratley[190], Leanne M. Redman[191], Maria J. Redondo[3,4], Rebecca M. Reynolds[59], Robert K. Semple[59,192], Jennifer L. Sherr[193], Emily K. Sims[2,38,39], Arianne Sweeting[194,195], Tiinamaija Tuomi[50,51,52], Miriam S. Udler[24,25,35,36], Kimberly K. Vesco[196], Tina Vilsbøll[197,198], Robert Wagner[69,70,199], Stephen S. Rich[115] & Paul W. Franks[22,26,107,113]

[21]Division of Preventive Medicine, Department of Medicine, Brigham and Women's Hospital and Harvard Medical School, Boston, MA, USA. [22]Department of Nutrition, Harvard T.H. Chan School of Public Health, Boston, MA, USA. [23]Novo Nordisk Foundation Center for Basic Metabolic Research, Faculty of Health and Medical Sciences, University of Copenhagen, Copenhagen, Denmark. [24]Diabetes Unit, Endocrine Division, Massachusetts General Hospital, Boston, MA, USA. [25]Center for Genomic Medicine, Massachusetts General Hospital, Boston, MA, USA. [26]Department of Clinical Sciences, Lund University Diabetes Centre, Lund University, Malmö, Sweden. [27]Department of Obstetrics and Gynaecology, The Rosie Hospital, Cambridge, UK. [28]NIHR Cambridge Biomedical Research Centre, University of Cambridge, Cambridge, UK. [29]Departments of Medicine and Community Health Sciences, Cumming School of Medicine, University of Calgary, Calgary, AB, Canada. [30]Department of Molecular Genetics, Madras Diabetes Research Foundation, Chennai, India. [31]Division of Pediatric Endocrinology, Department of Pediatrics, Saint Louis University School of Medicine, SSM Health Cardinal Glennon Children's Hospital, St. Louis, MO, USA. [32]CIBER-BBN, ISCIII, Madrid, Spain. [33]Institut d'Investigació Biomèdica Sant Pau (IIB SANT PAU), Barcelona, Spain. [34]Departament de Medicina, Universitat Autònoma de Barcelona, Bellaterra, Spain. [35]Programs in Metabolism and Medical & Population Genetics, Broad Institute, Cambridge, MA, USA. [36]Department of Medicine, Harvard Medical School, Boston, MA, USA. [37]Division of Endocrinology, Diabetes and Metabolism, Johns Hopkins University School of Medicine, Baltimore, MD, USA. [38]Department of Pediatrics, Indiana University School of Medicine, Indianapolis, IN, USA. [39]Center for Diabetes and Metabolic Diseases, Indiana University School of Medicine, Indianapolis, IN, USA. [40]Department of Biostatistics and Epidemiology, Rutgers School of Public Health, Piscataway, NJ, USA. [41]University Hospital Leuven, Leuven, Belgium. [42]Department of Nutrition, Université de Montréal, Montreal, QC, Canada. [43]Research Center, Sainte-Justine University Hospital Center, Montreal, QC, Canada. [44]Department of Pediatrics, Erasmus Medical Center, Rotterdam, The Netherlands. [45]Division of Population Health & Genomics, School of Medicine, University of Dundee, Dundee, UK. [46]Department of Pediatrics, Stanford School of Medicine, Stanford University, Stanford, CA, USA. [47]Stanford Diabetes Research Center, Stanford School of Medicine, Stanford University, Stanford, CA, USA. [48]University of Florida, Gainesville, FL, USA. [49]Department of Nutrition, University of North Carolina at Chapel Hill, Chapel Hill, NC, USA. [50]Helsinki University Hospital, Abdominal Centre/Endocrinology, Helsinki, Finland. [51]Folkhalsan Research Center, Helsinki, Finland. [52]Institute for Molecular Medicine Finland FIMM, University of Helsinki, Helsinki, Finland. [53]Department of Pediatrics, Division of Endocrinology, Boston Children's Hospital, Boston, MA, USA. [54]Department of Medicine, Faculty of Medicine, University of Malaya, Kuala Lumpur, Malaysia. [55]Asia Diabetes Foundation, Hong Kong SAR, China. [56]Department of Medicine & Therapeutics, Chinese University of Hong Kong, Hong Kong SAR, China. [57]Departments of Pediatrics and Clinical Genetics, Kuopio University Hospital, Kuopio, Finland. [58]Department of Medicine, University of Eastern Finland, Kuopio, Finland. [59]Centre for Cardiovascular Science, Queen's Medical Research Institute, University of Edinburgh, Edinburgh, UK. [60]Department of Epidemiology, University of Pittsburgh, Pittsburgh, PA, USA. [61]Metabolic Disease Unit, University Hospital of Padova, Padova, Italy. [62]Department of Medicine, University of Padova, Padova, Italy. [63]Department of Orthopedics, Zuyderland Medical Center, Sittard-Geleen, The Netherlands. [64]Departments of Pediatrics and Medicine, University of Chicago, Chicago, IL, USA. [65]Welch Center for Prevention, Epidemiology, and Clinical Research, Johns Hopkins Bloomberg School of Public Health, Baltimore, MD, USA. [66]Ciccarone Center for the Prevention of Cardiovascular Disease, Johns Hopkins School of Medicine, Baltimore, MD, USA. [67]Department of Medicine, Johns Hopkins University, Baltimore, MD, USA. [68]Department of Health Policy and Management, Johns Hopkins University Bloomberg School of Public Health, Baltimore, MD, USA. [69]Institute for Clinical Diabetology, German Diabetes Center, Leibniz Center for Diabetes Research at Heinrich Heine University Düsseldorf, Auf'm Hennekamp 65, 40225 Düsseldorf, Germany. [70]German Center for Diabetes Research (DZD), Ingolstädter Landstraße 1, 85764 Neuherberg, Germany. [71]Section of Academic Primary Care, US Department of Veterans Affairs Eastern Colorado Health Care System, Aurora, CO, USA. [72]Department of Medicine, University of Colorado School of Medicine, Aurora, CO, USA. [73]Department of Epidemiology, Johns Hopkins Bloomberg School of Public Health, Baltimore, MD, USA. [74]Institute of Experimental Endocrinology, Biomedical Research Center, Slovak Academy of Sciences, Bratislava, Slovakia. [75]Clinical and Translational Epidemiology Unit, Massachusetts General Hospital, Boston, MA, USA. [76]Mohn Center for Diabetes Precision Medicine, Department of Clinical Science, University of Bergen, Bergen, Norway. [77]Children and Youth Clinic, Haukeland University Hospital, Bergen, Norway. [78]Eastern Health Clinical School, Monash

University, Melbourne, VIC, Australia. [79]Laboratory for Molecular Epidemiology in Diabetes, Li Ka Shing Institute of Health Sciences, The Chinese University of Hong Kong, Hong Kong, China. [80]Hong Kong Institute of Diabetes and Obesity, The Chinese University of Hong Kong, Hong Kong, China. [81]Children's Nutrition Research Center, USDA/ARS, Houston, TX, USA. [82]Stanford University School of Medicine, Stanford, CA, USA. [83]Internal Medicine, University of Manitoba, Winnipeg, MB, Canada. [84]Department of Diabetology, APHP, Paris, France. [85]Sorbonne Université, INSERM, NutriOmic team, Paris, France. [86]Department of Nutrition, Dietetics and Food, Monash University, Melbourne, VIC, Australia. [87]Monash Centre for Health Research and Implementation, Monash University, Clayton, VIC, Australia. [88]Health Management Center, The Second Affiliated Hospital of Chongqing Medical University, Chongqing Medical University, Chongqing, China. [89]MRC/Wits Developmental Pathways for Health Research Unit, Department of Paediatrics, Faculty of Health Sciences, University of the Witwatersrand, Johannesburg, South Africa. [90]Channing Division of Network Medicine, Brigham and Women's Hospital, Boston, MA, USA. [91]Sydney Brenner Institute for Molecular Bioscience, Faculty of Health Sciences, University of the Witwatersrand, Johannesburg, South Africa. [92]Department of Women and Children's health, King's College London, London, UK. [93]Life-course Epidemiology of Adiposity and Diabetes (LEAD) Center, University of Colorado Anschutz Medical Campus, Aurora, CO, USA. [94]Section of Adult and Pediatric Endocrinology, Diabetes and Metabolism, Kovler Diabetes Center, University of Chicago, Chicago, IL, USA. [95]Department of Pediatrics, Riley Hospital for Children, Indiana University School of Medicine, Indianapolis, IN, USA. [96]Biomedical Research Institute Girona, IdIBGi, Girona, Spain. [97]Diabetes, Endocrinology and Nutrition Unit Girona, University Hospital Dr Josep Trueta, Girona, Spain. [98]Institute of Health System Science, Feinstein Institutes for Medical Research, Northwell Health, Manhasset, NY, USA. [99]University of California at San Francisco, Department of Pediatrics, Diabetes Center, San Francisco, CA, USA. [100]Division of Endocrinology, Diabetes and Metabolism, Cedars-Sinai Medical Center, Los Angeles, CA, USA. [101]Department of Medicine, Cedars-Sinai Medical Center, Los Angeles, CA, USA. [102]Adelaide Medical School, Faculty of Health and Medical Sciences, The University of Adelaide, Adelaide, SA, Australia. [103]Robinson Research Institute, The University of Adelaide, Adelaide, SA, Australia. [104]Department of Public Health and Novo Nordisk Foundation Center for Basic Metabolic Research, Faculty of Health and Medical Sciences, University of Copenhagen, 1014 Copenhagen, Denmark. [105]Division of Endocrinology and Diabetes, Department of Pediatrics, Sanford Children's Hospital, Sioux Falls, SD, USA. [106]University of South Dakota School of Medicine, E Clark St, Vermillion, SD, USA. [107]Oxford Centre for Diabetes, Endocrinology and Metabolism, University of Oxford, Oxford, UK. [108]Division of General Internal Medicine, Massachusetts General Hospital, Boston, MA, USA. [109]UPMC Children's Hospital of Pittsburgh, Pittsburgh, PA, USA. [110]Department of Medicine, Northwestern University Feinberg School of Medicine, Chicago, IL, USA. [111]Department of Pathology & Molecular Medicine, McMaster University, Hamilton, ON, Canada. [112]Population Health Research Institute, Hamilton, ON, Canada. [113]Department of Translational Medicine, Medical Science, Novo Nordisk Foundation, Tuborg Havnevej 19, 2900 Hellerup, Denmark. [114]Department of Diabetes and Endocrinology, Nelson R Mandela School of Medicine, University of KwaZulu-Natal, Durban, South Africa. [115]Center for Public Health Genomics, Department of Public Health Sciences, University of Virginia, Charlottesville, VA, USA. [116]Division of Epidemiology and Community Health, School of Public Health, University of Minnesota, Minneapolis, MN, USA. [117]Department of Chronic Diseases and Metabolism, Clinical and Experimental Endocrinology, KU Leuven, Leuven, Belgium. [118]School of Health and Wellbeing, College of Medical, Veterinary and Life Sciences, University of Glasgow, Glasgow, UK. [119]Department of Obstetrics, Gynecology, and Reproductive Biology, Massachusetts General Hospital and Harvard Medical School, Boston, MA, USA. [120]Sanford Children's Specialty Clinic, Sioux Falls, SD, USA. [121]Department of Biostatistics, Johns Hopkins Bloomberg School of Public Health, Baltimore, MD, USA. [122]Centre for Physical Activity Research, Rigshospitalet, Copenhagen, Denmark. [123]Institute for Sports and Clinical Biomechanics, University of Southern Denmark, Odense, Denmark. [124]Department of Medicine, Division of Endocrinology, Diabetes and Metabolism, Indiana University School of Medicine, Indianapolis, IN, USA. [125]AMAN Hospital, Doha, Qatar. [126]Department of Preventive Medicine, Division of Biostatistics, Northwestern University Feinberg School of Medicine, Chicago, IL, USA. [127]Institute of Molecular and Genomic Medicine, National Health Research Institutes, Zhunan, Taiwan. [128]Division of Endocrinology and Metabolism, Taichung Veterans General Hospital, Taichung, Taiwan. [129]Division of Endocrinology and Metabolism, Taipei Veterans General Hospital, Taipei, Taiwan. [130]Barbara Davis Center for Diabetes, University of Colorado Anschutz Medical Campus, Aurora, CO, USA. [131]University Hospital of Tübingen, Tübingen, Germany. [132]Institute of Diabetes Research and Metabolic Diseases (IDM), Helmholtz Center Munich, Neuherberg, Germany. [133]Steno Diabetes Center Aarhus, Aarhus University Hospital, Aarhus, Denmark. [134]University of Newcastle, Newcastle upon Tyne, UK. [135]Sections on Genetics and Epidemiology, Joslin Diabetes Center, Harvard Medical School, Boston, MA, USA. [136]Department of Clinical Pharmacy and Pharmacology, University Medical Center Groningen, Groningen, The Netherlands. [137]Gastroenterology, Baylor College of Medicine, Houston, TX, USA. [138]Department of Endocrinology, University Hospitals Leuven, Leuven, Belgium. [139]Sorbonne University, Inserm U938, Saint-Antoine Research Centre, Institute of Cardiometabolism and Nutrition, 75012 Paris, France. [140]Department of Endocrinology, Diabetology and Reproductive Endocrinology, Assistance Publique-Hôpitaux de Paris, Saint-Antoine University Hospital, National Reference Center for Rare Diseases of Insulin Secretion and Insulin Sensitivity (PRISIS), Paris, France. [141]Deakin University, Melbourne, VIC, Australia. [142]Department of Epidemiology, Madras Diabetes Research Foundation, Chennai, India. [143]Department of Diabetes and Endocrinology, Guy's and St Thomas' Hospitals NHS Foundation Trust, London, UK. [144]School of Agriculture, Food and Wine, University of Adelaide, Adelaide, SA, Australia. [145]Institut Cochin, Inserm U, 10116 Paris, France. [146]Pediatric Endocrinology and Diabetes, Hopital Necker Enfants Malades, APHP Centre, Université de Paris, Paris, France. [147]Department of Medical Genetics, Haukeland University Hospital, Bergen, Norway. [148]Department of Medicine, University of Maryland School of Medicine, Baltimore, MD, USA. [149]Department of Epidemiology, Geisel School of Medicine at Dartmouth, Hanover, NH, USA. [150]Nephrology, Dialysis and Renal Transplant Unit, IRCCS—Azienda Ospedaliero-Universitaria di Bologna, Alma Mater Studiorum University of Bologna, Bologna, Italy. [151]Department of Medical Genetics, AP-HP Pitié-Salpêtrière Hospital, Sorbonne University, Paris, France. [152]Global Center for Asian Women's Health, Yong Loo Lin School of Medicine, National University of Singapore, Singapore, Singapore. [153]Department of Obstetrics and Gynecology, Yong Loo Lin School of Medicine, National University of Singapore, Singapore, Singapore. [154]Kaiser Permanente Northern California Division of Research, Oakland, CA, USA. [155]Department of Epidemiology and Biostatistics, University of California San Francisco, San Francisco, CA, USA. [156]National Institute of Diabetes and Digestive and Kidney Diseases, National Institutes of Health, Bethesda, MD, USA. [157]Department of Health Research Methods, Evidence, and Impact, Faculty of Health Sciences, McMaster University, Hamilton, ON, Canada. [158]Ann & Robert H. Lurie Children's Hospital of Chicago, Department of Pediatrics, Northwestern University Feinberg School of Medicine, Chicago, IL, USA. [159]Department of Clinical and Organizational Development, Chicago, IL, USA. [160]American Diabetes Association, Arlington, VA, USA. [161]College of Medicine and Health Sciences, University of Gondar, Gondar, Ethiopia. [162]Global Health Institute, Faculty of Medicine and Health Sciences, University of Antwerp, 2160 Antwerp, Belgium. [163]Department of Medicine and Kovler Diabetes Center, University of Chicago, Chicago, IL, USA. [164]School of Nursing, Faculty of Health Sciences, McMaster University, Hamilton, ON, Canada. [165]Division of Endocrinology, Metabolism, Diabetes, University of Colorado Anschutz Medical Campus, Aurora, CO, USA. [166]Department of Clinical Medicine, School of Medicine, Trinity College Dublin, Dublin, Ireland. [167]Department of Endocrinology, Wexford General Hospital, Wexford, Ireland. [168]Division of Endocrinology, NorthShore University HealthSystem, Skokie, IL, USA. [169]Department of Medicine, Pritzker School of Medicine, University of Chicago, Chicago, IL, USA. [170]Department of Genetics, Stanford School of Medicine, Stanford University, Stanford, CA, USA. [171]Faculty of Health, Aarhus University, Aarhus, Denmark. [172]Departments of Pediatrics and Medicine and Kovler Diabetes Center, University of Chicago, Chicago, IL, USA. [173]University of Washington School of Medicine, Seattle, WA, USA. [174]Department of Population Medicine, Harvard Medical School, Harvard Pilgrim Health Care Institute, Boston, MA, USA. [175]Department of Medicine, Universite de Sherbrooke, Sherbrooke, QC, Canada. [176]Department of Internal Medicine, Seoul National University College of Medicine, Seoul National University Hospital, Seoul, Republic of Korea. [177]Joslin Diabetes Center, Harvard Medical School, Boston, MA, USA. [178]Charles Bronfman Institute for Personalized Medicine, Icahn School of Medicine at Mount Sinai, New York, NY, USA. [179]Broad Institute, Cambridge, MA, USA. [180]Division of Metabolism, Digestion and Reproduction, Imperial College London, London, UK. [181]Department of

Diabetes & Endocrinology, Imperial College Healthcare NHS Trust, London, UK. [182]Department of Diabetology, Madras Diabetes Research Foundation & Dr. Mohan's Diabetes Specialities Centre, Chennai, India. [183]Department of Medicine, Faculty of Medicine and Health Sciences, University of Auckland, Auckland, New Zealand. [184]Auckland Diabetes Centre, Te Whatu Ora Health New Zealand, Auckland, New Zealand. [185]Medical Bariatric Service, Te Whatu Ora Counties, Health New Zealand, Auckland, New Zealand. [186]Oxford NIHR Biomedical Research Centre, University of Oxford, Oxford, UK. [187]University of Cambridge, Metabolic Research Laboratories and MRC Metabolic Diseases Unit, Wellcome-MRC Institute of Metabolic Science, Cambridge, UK. [188]Department of Epidemiology & Public Health, University of Maryland School of Medicine, Baltimore, MD, USA. [189]Department of Internal Medicine, Division of Metabolism, Endocrinology and Diabetes, University of Michigan, Ann Arbor, MI, USA. [190]AdventHealth Translational Research Institute, Orlando, FL, USA. [191]Pennington Biomedical Research Center, Baton Rouge, LA, USA. [192]MRC Human Genetics Unit, Institute of Genetics and Cancer, University of Edinburgh, Edinburgh, UK. [193]Yale School of Medicine, New Haven, CT, USA. [194]Faculty of Medicine and Health, University of Sydney, Sydney, NSW, Australia. [195]Department of Endocrinology, Royal Prince Alfred Hospital, Sydney, NSW, Australia. [196]Kaiser Permanente Northwest, Kaiser Permanente Center for Health Research, Portland, OR, USA. [197]Clinial Research, Steno Diabetes Center Copenhagen, Herlev, Denmark. [198]Department of Clinical Medicine, Faculty of Health and Medical Sciences, University of Copenhagen, Copenhagen, Denmark. [199]Department of Endocrinology and Diabetology, University Hospital Düsseldorf, Heinrich Heine University Düsseldorf, Moorenstr. 5, 40225 Düsseldorf, Germany.

