## [Peer Review File · Communications Medicine]

Reviewers' comments:

Reviewer #1 (Remarks to the Author):

The manuscript by Felton et al. is a well-written literature review on papers published in 2011-2022 and focusing on islet autoantibodies in T1D. The authors have taken a very challenging task to summarize the significance of islet autoantibodies in four occurrences: 1) before diagnosis, 2) at diagnosis, 3) within a year after diagnosis, and even 4) in the context of clinical trials. The authors have managed to review a large number of articles on these topics and have created a summary of the findings. However, the majority of published papers on islet autoantibodies deals with 1). This is reflected on the balance of this literature review and the conclusions that strongly focus on 1).

Several questions remain and improvements are needed:

The decision to focus on publication years 2011-2022 is understandable because of the vast number of publications on islet autoantibodies, but on the other hand this restriction leaves out important earlier papers. Also, some important recent papers concerning 1) should still be added. In particular, earlier attempts to combine and harmonize data from several studies should be better emphasized (e.g. >10 papers published by the T1DI Consortium with >24,000 prospectively observed before development of islet autoimmunity and during progression from autoimmunity to clinical diagnosis of T1D). Analyses from a very large combined cohort most reliably reveal the key aspects of disease heterogeneity compared to small individual cohorts. 2) and 3) have been analyzed in numerous studies since the discovery of islet autoantibodies. It does not seem reasonable to include ALL studies published in 2011-2022, small and large ones, and not to include any earlier studies. A better and more comprehensive view would be obtained by including also older studies, at least those with large sample size. 4) is questionable to present in the review because inclusion criteria of many clinical trials have included a definitive islet autoantibody status and therefore nothing can be concluded about the role of interventions in individuals with other kind of islet autoantibody profiles. Furthermore, many intervention trials have performed posthoc subgroup analyses between participants with distinct autoantibody profiles, and the results reporting about different effects of the intervention in these groups are only suggestive (maybe false!) and should be confirmed in a trial specifically designed for these subgroups. The authors should acknowledge this in the Discussion as a limitation.

Introduction

Line 103: reference missing

Line 105: "studies and clinical trials" should be replaced by: prospective and cross-sectional studies

Line 111: ref 5-8 are old, please, include the most recent references from DASP/IASP workshops

Line 113: "hypothesized that islet autoantibodies" should be replaced by: explored and summarized the evidence how islet autoantibodies... It has been reported and confirmed in many studies that the disease process resulting in clinical T1D is heterogeneous.

Methods.

Why Lund University librarians in Sweden perform the literature search? The study group includes investigators from UK, Australia and mostly from the US, but none from Sweden.

Line 138: what is meant by " ≥ 10 /group"? Which group?

It is difficult to fit the text in the Methods with Figure 1. For example, the text indicates that "several key articles identified by the group were included" but this is not shown in Figure 1. This methodology does not sound like a systematic review but rather an expert opinion. Is the review a combination of systematic review and expert opinion? If yes, this should be clearly stated in the Abstract, Methods, Results, Discussion.

It is unclear whether LADA was included in the search criteria or not? LADA is considered a subtype

of T1D and should have been included.

Lines 165-167: Please, give reasoning why this checklist was selected and needed.

Results.

Tables 1 and 2 are not really central for the Results. No clear conclusions could be drawn from the 11 studies presented in Table 1. Instead, Supplementary Table 1 (which is too big to be included in the Main text) is the most informative for the conclusions of this review. The authors should extract the most important findings from Supplementary table 1, that have been repeated in several studies, and create a new Table to be included in the Main text.

Table 3 just repeats the text and does not give additional information. Can be removed.

Table 4 is a problem. It contains several mistakes in classification of the studies. In fact, there is no difference between birth cohort studies and longitudinal studies. Birth cohort studies have followed individuals longitudinally from birth (ABIS, BABYDIAB, BABYDIET, DAISY, DIPP, DiPiS, Fr1DA, TEDDY). Other longitudinal studies have recruited family members and started longitudinal follow-up from any age (BDR, DPT-1, TrialNet). FPDR is a cross-sectional study from the time of stage 3 T1D of cases and their FDRs.

Line 230: remove "Since then", because many studies reported similar results before ref.30.

Lines 330-337: Information on the performance of other methods in addition to RBA and ECL should be added.

Lines 344-346: Please, add data clearly reporting that IAA is much less often positive in adults compared to children

How did the authors (and the reviewed studies) take into account the fact that after diagnosis patients will develop insulin antibodies as a consequence of insulin treatment?

Discussion.

Line 428: replace "hypothesized" as suggested above.

Line 430: revise "systematically" because also papers identified by the experts were included

Line 433: remove "Strikingly"

Line 432; "past 10 years" revise to reflect the revised manuscript

Lines 453-454: replace "our analysis" with "our literature review"

Line 465: replace "trials" with "studies"

Lines 487-500: The authors should comment whether it is relevant to follow islet autoantibodies after diagnosis? If yes, why? If not, why?

Lines 544-550: why discussion of the Fr1da study was added? Should be removed or alternatively, extend the discussion to include also other studies not reviewed.

Lines 548-550: repeats the text in the Methods, can be removed.

Suggestion: the authors should consider performing a meta-analysis of studies prior diagnosis of stage 3 T1D. This should be possible. Can at least be discussed.

References:

Please, remove duplicates.

Please, add new references as suggested above (T1DI, DASP/IASP, older studies with large sample sizes)

Reviewer #2 (Remarks to the Author):

This T1D Diagnostics-focused Systematic Review written on behalf of the ADA/EASD PMDI performed to ascertain whether islet autoantibodies, biomarkers of autoimmunity in the pancreas, could aid in stratifying individuals with different clinical presentations of T1D. Then, they found existing evidence most strongly supporting the application of these biomarkers to the period before

clinical diagnosis, when certain autoantibody features (number, type) and the age when autoantibodies first develop.

The review includes 151 papers (a huge amount of data analyzed) published over the past 10 years meeting inclusion criteria and identified recurring themes in the literature. The method used is correct and the bias of different studies has been adjusted. And the writing is extremely fluent. It is a valuable systematic review which could provide important information for patients and care providers on what to expect for future type 1 diabetes progression.

The findings support continued use of pre-clinical staging paradigms based on autoantibody number and suggest that additional autoantibody features, particularly in relation to age and genetic risk, could offer more precise stratification. What's more, this review implies that the prediction, diagnosis and treatment of diabetes has totally entered the era of precision medicine.

Quite a few changes need to be made and questions to be answered:

1. As it described, only forty-four percent of total papers 403 (65/151) referenced participation in an autoantibody standardization program, could it possible to list the specific detection methods of islet autoantibodies in different studies and label the lab whether in above program or not in supplementary data?
2. In the primary result, islet autoantibodies are likely to be most useful to define T1D heterogeneity prior to clinical diagnosis, is there any study identifying two profiles, such as autoantibody number in combination with its titer or affinity to predict the stage 3 T1D progression?
3. Only 10/151 (7%) studies focused on a population that did not feature primarily European ancestry, thus, could it possible to list the population in different studies in supplementary data?

Reviewers' comments:

Reviewer #1 (Remarks to the Author):

The manuscript by Felton et al. is a well-written literature review on papers published in 2011-2022 and focusing on islet autoantibodies in T1D. The authors have taken a very challenging task to summarize the significance of islet autoantibodies in four occurrences: 1) before diagnosis, 2) at diagnosis, 3) within a year after diagnosis, and even 4) in the context of clinical trials. The authors have managed to review a large number of articles on these topics and have created a summary of the findings. However, the majority of published papers on islet autoantibodies deals with 1). This is reflected on the balance of this literature review and the conclusions that strongly focus on 1).

Several questions remain and improvements are needed:

The decision to focus on publication years 2011-2022 is understandable because of the vast number of publications on islet autoantibodies, but on the other hand this restriction leaves out important earlier papers. Also, some important recent papers concerning 1) should still be added. In particular, earlier attempts to combine and harmonize data from several studies should be better emphasized (e.g. >10 papers published by the T1DI Consortium with >24,000 prospectively observed before development of islet autoimmunity and during progression from autoimmunity to clinical diagnosis of T1D). Analyses from a very large combined cohort most reliably reveal the key aspects of disease heterogeneity compared to small individual cohorts. 2) and 3) have been analyzed in numerous studies since the discovery of islet autoantibodies. It does not seem reasonable to include ALL studies published in 2011-2022, small and large ones, and not to include any earlier studies. A better and more comprehensive view would be obtained by including also older studies, at least those with large sample size.

We thank the reviewer for the close reading of this manuscript and agree that the number of publications on islet autoantibodies is vast and heterogenous. We agree that by limiting the date range, inevitably, some important papers will be excluded from the search. As part of the larger Precision Medicine in Diabetes Initiative, the published PROSPERO search protocol was developed a priori in order to maximize results and feasibility. In fact, our initial search did include a broader date range; however, this yielded over 60,000 papers, analysis of which was not feasible with our available resources. Based on this, and after input from the editors, we decided to not revise the search strategy and PROSPERO protocol.

However, we agree with the reviewer that analysis from the large T1DI Consortium provides important insight into disease heterogeneity based on autoantibody positivity. In addition to the T1DI papers that were already captured by the search and included in the review, the manuscript by Anand et al (Diabetes Care 2021) has now been included, and additional emphasis on lessons from the T1DI cohort that were published outside of the date range, but include important and significant findings, are now highlighted in the discussion (lines 497-500, 570-577).

4) is questionable to present in the review because inclusion criteria of many clinical trials have included a definitive islet autoantibody status and therefore nothing can be concluded about the role of interventions in individuals with other kind of islet autoantibody profiles. Furthermore, many intervention trials have performed posthoc subgroup analyses between participants with distinct

autoantibody profiles, and the results reporting about different effects of the intervention in these groups are only suggestive (maybe false!) and should be confirmed in a trial specifically designed for these subgroups. The authors should acknowledge this in the Discussion as a limitation.

While we understand the reviewer's perspective, we feel strongly that a potential link to treatment response is an important piece of the definition of T1D endotypes, as the ultimate goal in understanding T1D heterogeneity is identification of individuals who have the most potential to benefit from specific treatments. We agree with the reviewer that post-hoc analysis is suggestive, but not sufficient, and that application of findings revealed in post-hoc analysis will require development of prospective trials with prespecified hypotheses. This is certainly a limitation of this data. However, an important component of the systematic review process is systematic assessment of the current state of the data, including quality and risks of bias. For these reasons, inclusion of analyses linking specific autoantibody characteristics (and the weaknesses in these analyses) is key to include as part of this review.

To ensure we have made this position (which we share with the reviewer) clear, our discussion includes a call for prospective testing of these hypotheses, which has now been revised to further emphasize its importance (lines 549-551, 583-586). As highlighted, an important conclusion of this review is that the quality of existing evidence limits its application to precision medicine.

In addition, we have revised the wording of the statement describing the three trials identified that limited testing to individuals positive for specific autoantibodies (lines 383-384). Of note, none of the trials that limited inclusion criteria to specific autoantibody positive individuals reported positive findings.

Introduction

Line 103: reference missing

At the time of original submission, the consensus statement was under review, but has now been accepted for publication in Nature Medicine. This is now noted in the text (lines 103-105).

Line 105: "studies and clinical trials" should be replaced by: prospective and cross-sectional studies

Because some studies included in this review did not fall into the prospective or cross-sectional studies classification (for example, retrospective analysis of observational cohorts), and because we felt it was important to include the distinction of clinical trials (vs studies), as this is an important component of our analysis, we chose to address this comment by revising this phrase to "observational studies and clinical trials" (line 107).

Line 111: ref 5-8 are old, please, include the most recent references from DASP/IASP workshops

We have now included a more recent reference that compares results from the 2018 and 2020 workshops (Marzinotto et al, Diabetologia 2023). Results from the 2023 IASP have not yet been published.

Line 113: "hypothesized that islet autoantibodies" should be replaced by: explored and summarized the evidence how islet autoantibodies... It has been reported and confirmed in many studies that the disease process resulting in clinical T1D is heterogeneous.

This sentence has been revised as suggested: “We explored and summarized evidence that islet autoantibodies can be used to identify unique phenotypes of disease progression...” (line 116).

Methods.

Why Lund University librarians in Sweden perform the literature search? The study group includes investigators from UK, Australia and mostly from the US, but none from Sweden.

This review was published as part of the larger Precision Medicine in Diabetes Initiative (PMDI), whose co-Chair, Paul Franks, has an academic appointment at Lund University in Sweden. As Lund University provided Covidence support with trained librarians, the PMDI Working Groups all used Lund-sponsored Covidence software with Prospero registration and Lund University librarians for the evidence-based systematic reviews. The Lund librarians are recognized in the “Acknowledgements” section, as recommended by PMDI leadership.

Line 138: what is meant by “≥ 10/group”? Which group?

This sentence has been revised to clarify that studies needed to include at least 10 participants in each arm of the study: “Studies must have had a total sample size ≥ 10 per experimental or control group studied...” (line 141).

It is difficult to fit the text in the Methods with Figure 1. For example, the text indicates that “several key articles identified by the group were included” but this is not shown in Figure 1. This methodology does not sound like a systematic review but rather an expert opinion. Is the review a combination of systematic review and expert opinion? If yes, this should be clearly stated in the Abstract, Methods, Results, Discussion.

Our search identified 11,183 papers via PubMed and EMBASE. Of these, 143 met inclusion criteria after full text screening. At that point, an additional 8 papers that met inclusion criteria but were not included in the search were identified by our group of experts. In response to this review, we have added 1 additional paper from the T1DI cohort, to make the total of additional papers identified 9. To address this comment, Figure 1 has now been revised to specifically reflect these numbers.

We respectfully disagree with the reviewer’s concern that our systematic review does not meet the criteria for a systematic review based on inclusion of articles identified by experts outside of the search we performed. A systematic review is not defined by the search approach, but rather the systematic application of prospectively defined inclusion/exclusion criteria to selected articles and the systematic extraction and analysis of the findings and their validity/risk of bias, with a systematic presentation of their findings (www.cochrane.org). Although a very small percentage of articles we screened (8/11191) were identified outside of our PubMed and Embase searches, these articles were subject to the same procedures as articles identified via our search. In fact, it is standard procedure to identify eligible articles that fall outside of the initial search as part of the systematic review process. As long as additional articles are scrutinized for inclusion/exclusion screening like any other paper (as these 8 papers were), there should not be a concern.

It is unclear whether LADA was included in the search criteria or not? LADA is considered a subtype of T1D and should have been included.

We agree that LADA could be considered a subtype of T1D; however, upon initial reading of the literature, we found that LADA was inconsistently defined, and frequently included participants that may have type 2 diabetes. Therefore, we made the decision to exclude these patients from the current review, as stated in the text in lines (143-146). However, we agree with the reviewer that this is a population of great interest for future studies, and have added comments regarding this limitation in the discussion (lines 573-577).

Lines 165-167: Please, give reasoning why this checklist was selected and needed.

Critical appraisal and quality assessment are necessary and inherent features of the systematic review process (<https://onlinelibrary.wiley.com/doi/epdf/10.1111/jebm.12141>). The Joanna Briggs Institute Critical Appraisal Tools are publicly available and widely used for systematic review quality assessment. The decision to use the JBI critical appraisal checklist was made with support from PMDI leadership with expertise in quality assessment.

Results.

Tables 1 and 2 are not really central for the Results. No clear conclusions could be drawn from the 11 studies presented in Table 1. Instead, Supplementary Table 1 (which is too big to be included in the Main text) is the most informative for the conclusions of this review. The authors should extract the most important findings from Supplementary table 1, that have been repeated in several studies, and create a new Table to be included in the Main text.

We agree that Supplementary Table 1 is the most informative table, given the number of papers identified that address the period prior to diagnosis. And, as the reviewer has pointed out, it is unfortunately too large to be included in the main text. As suggested, we have now added a table summarizing key findings from papers analyzing the period prior to diagnosis that is small enough to be included in the main text (Table 1).

While papers in tables 1 and 2 do not lead to clear conclusions, an important component of systematic review is systematic extraction and presentation of summary data. Additionally, these tables provide a useful reference of papers for the corresponding clinical periods that have been systematically reviewed and summarized. Because their size permits, we have decided to leave these tables in the main text (they are now tables 2 and 3).

Table 3 just repeats the text and does not give additional information. Can be removed.

We included table 3 as a quick reference for definitions of the time periods studied, and to further emphasize that results are heavily weighted in the "prior to diagnosis" category. Based on this comment, this table has now been condensed and incorporated into Figure 1 to highlight the distribution of papers into each group.

Table 4 is a problem. It contains several mistakes in classification of the studies. In fact, there is no

difference between birth cohort studies and longitudinal studies. Birth cohort studies have followed individuals longitudinally from birth (ABIS, BABYDIAB, BABYDIET, DAISY, DIPP, DiPiS, Fr1DA, TEDDY).

Other longitudinal studies have recruited family members and started longitudinal follow-up from any age (BDR, DPT-1, TrialNet). FPDR is a cross-sectional study from the time of stage 3 T1D of cases and their FDRs.

We thank the reviewer for the suggestions and have revised this table to more specifically and correctly characterize each cohort.

Line 230: remove “Since then”, because many studies reported similar results before ref.30.

This sentence has been revised as suggested: “This appreciation that lifetime risk of diabetes progression nears 100% once multiple positive islet autoantibodies have developed informs the current T1D staging system 2, and the impact of autoantibody number on risk of progression to stage 3 has been corroborated in numerous studies of additional cohorts” (line 233).

Lines 330-337: Information on the performance of other methods in addition to RBA and ECL should be added.

While papers that used other methods to measure autoantibodies were included in the search results, none, with the exception of the papers that compared RBA and ECL assays, specifically tested the performance of the assay used against others, or considered the assay type as a source of heterogeneity. Because of this, the reference focused on the RBA vs ECL comparison. In response to this comment a sentence has now been added to recognize that these were not the only 2 assays used (lines 333-336). In addition, Tables 2 and 3, and Supplemental Tables 1 and 2 have also been revised to include which assays were described in each paper.

Lines 344-346: Please, add data clearly reporting that IAA is much less often positive in adults compared to children

We agree that this is important data to highlight and have now added lines 355-357.

How did the authors (and the reviewed studies) take into account the fact that after diagnosis patients will develop insulin antibodies as a consequence of insulin treatment?

Thank you for highlighting this need for clarification. Autoantibody studies used to characterize progression after diagnosis or in response to treatment were antibodies obtained at the time of diagnosis, and prior to treatment (this confirmation was included in the extraction step of the systematic review). This means that individuals who were positive for IAA were positive for IAA prior to initiation of insulin. This has been clarified in the text (lines 364-366).

Discussion.

Line 428: replace “hypothesized” as suggested above.

This has been revised as suggested (line 440).

Line 430: revise “systematically” because also papers identified by the experts were included

As noted above, we respectfully disagree that the inclusion of papers identified by experts disqualifies our work as a systematic review. The prospectively defined inclusion and exclusion criteria were applied to all selected papers. All papers were subject to the same systematic extraction of data, analysis of findings and validity/risk of bias, regardless of how these papers were identified (www.cochrane.org).

Line 433: remove “Strikingly”

This has been removed as suggested.

Line 432; “past 10 years” revise to reflect the revised manuscript

While the search criteria have not been changed, we have now included additional discussion mentioning that important papers that fell outside of our review window were excluded with specific examples in the discussion (lines 497-500, 570-577).

Lines 453-454: replace “our analysis” with “our literature review”

As described above, we feel that this work meets the standards of a systematic review and have kept the original language.

Line 465: replace “trials” with “studies”

This has been revised as suggested.

Lines 487-500: The authors should comment whether it is relevant to follow islet autoantibodies after diagnosis? If yes, why? If not, why?

While this is an interesting question, as part of the Precision Medicine Initiative, we were specifically tasked with addressing precision medicine surrounding T1D diagnosis. Based on the initial consensus report, this involves “refining the characterization of the diabetes diagnosis for therapeutic optimization or to improve prognostic clarity using information about a person’s unique biology, environment, and/or context” (Chung et al. Diabetes Care 2021). Therefore, whether it is relevant to continue to follow islet autoantibodies after diagnosis falls outside the scope of this review.

Lines 544-550: why discussion of the Fr1da study was added? Should be removed or alternatively, extend the discussion to include also other studies not reviewed.

Because we agree with the reviewer’s initial point that some important papers were either published before our search criteria or after the search, we wanted to highlight this limitation with examples of relevant papers. Discussion of the Fr1da study was included as example of a relevant study that was not included because it fell outside the date range of our search criteria. As suggested by the reviewer, we

have now also included discussion of additional papers from the T1DI initiative that fell outside of the search window (lines 497-500, 570-577). We also referenced this limitation of our search strategy (lines 463-465); however, we felt that an inclusive list of all potential studies that were not included in the search was beyond the scope of our discussion.

Lines 548-550: repeats the text in the Methods, can be removed.

We have revised this line in response to the next comment.

Suggestion: the authors should consider performing a meta-analysis of studies prior diagnosis of stage 3 T1D. This should be possible. Can at least be discussed.

While we did consider meta-analysis of studies at each timepoint, the vast heterogeneity of study outcomes, exposures, and study conditions prohibited this without harmonization of values and significant technical and statistical support that extended beyond the reach of this project. We have described this more clearly in lines 559-562. We have also highlighted the importance of the massive T1DI effort as an example of the work necessary for the harmonization and reanalysis of combined cohorts (lines 570-573).

References:

Please, remove duplicates.

Duplicates have been removed.

Please, add new references as suggested above (T1DI, DASP/IASP, older studies with large sample sizes)

As noted above, the recently published IASP/DASP paper has been added, an additional T1DI paper has been added to search results, extracted, and is now included. Additional papers, including other more recent T1DI papers that fell outside of our search window are now discussed as described above.

Reviewer #2 (Remarks to the Author):

This T1D Diagnostics-focused Systematic Review written on behalf of the ADA/EASD PMDI performed to ascertain whether islet autoantibodies, biomarkers of autoimmunity in the pancreas, could aid in stratifying individuals with different clinical presentations of T1D. Then, they found existing evidence most strongly supporting the application of these biomarkers to the period before clinical diagnosis, when certain autoantibody features (number, type) and the age when autoantibodies first develop.

The review includes 151 papers (a huge amount of data analyzed) published over the past 10 years meeting inclusion criteria and identified recurring themes in the literature. The method used is correct and the bias of different studies has been adjusted. And the writing is extremely fluent. It is a valuable systematic review which could provide important information for patients and care providers on what to expect for future type 1 diabetes progression.

The findings support continued use of pre-clinical staging paradigms based on autoantibody number and suggest that additional autoantibody features, particularly in relation to age and genetic risk, could offer more precise stratification. What's more, this review implies that the prediction, diagnosis and treatment of diabetes has totally entered the era of precision medicine.

We thank Reviewer 2 for the critique, including the recognition of the large amount of articles reviewed, acknowledgement that our systematic review methodology is sound, and comments on the review's value.

Quite a few changes need to be made and questions to be answered:

1. As it described, only forty-four percent of total papers 403 (65/151) referenced participation in an autoantibody standardization program, could it possible to list the specific detection methods of islet autoantibodies in different studies and label the lab whether in above program or not in supplementary data?

We agree that antibody standardization is a critical step to moving toward the application of precision medicine. In order to highlight the current heterogeneity, we have now added the assays described and whether assays are noted to have been tested in a standardization program in Tables 2 and 3 (main text), and Supplemental Tables 1 and 2.

2. In the primary result, islet autoantibodies are likely to be most useful to define T1D heterogeneity prior to clinical diagnosis, is there any study identifying two profiles, such as autoantibody number in combination with its titer or affinity to predict the stage 3 T1D progression?

We agree that a combination of antibody features will likely to aid T1D risk assessment, and we were encouraged that our data supports the concept of the use of additional features, in combination with autoantibody number, as elegantly reviewed in reference 65. Almost all studies assessed an autoantibody feature, in addition to number, as summarized in Supplemental Table 1. In response to the reviewer comment, we have enhanced our discussion of this concept in the discussion in lines 460-462.

3. Only 10/151 (7%) studies focused on a population that did not feature primarily European ancestry, thus, could it possible to list the population in different studies in supplementary data?

This has been added as Supplementary Table 3.

Reviewers' comments:

Reviewer #2 (Remarks to the Author):

As to the question "Is it possible to list the specific detection methods of islet autoantibodies in different studies and label the lab whether in above program or not in supplementary data?", the author added the assays described and whether assays were noted to have been tested in a standardization program in Tables 2 and 3 (main text), and Supplemental Tables 1 and 2.

As to the question "Is there any study identifying two profiles, such as autoantibody number in combination with its titer or affinity to predict the stage 3 T1D progression?", the author answered almost all studies assessed an autoantibody feature, in addition to number, as summarized in Supplemental Table 1. In response to the reviewer comment, they have enhanced their discussion of this concept in the discussion in lines 460-462.

As to the question "Is it possible to list the population in different studies in supplementary data?", this has been added as Supplementary Table 3.

After revision, I believe it is a valuable systematic review which could provide important information for patients and care providers on what to expect for future type 1 diabetes progression. I recommend it to be accepted.

I'm glad to provide help to review the manuscript and rebuttal letter again, and give some advices. I think the Reviewer 1's concerns have been suitably addressed by the authors, and I have some additional comments as follows:

1. Could the author label and note the 9 key articles identified by the group of experts that also met inclusion criteria in the method part?
2. As to the exclusion of LADA, I think author should also simply explain in method part, and should state in the discussion if the main results of this paper could also replicate in some of LADA studies instead of future study/review, apart from explaining the reason exclusion of LADA.

Reviewer #3 (Remarks to the Author):

This is a comprehensive and informative review article. The topic is timely and important.

Overall, the authors have adequately addressed the reviewers' comments.

The limited analysis of publications due to the time window 2011-2022 nevertheless remains a certain problem, since - as criticised by reviewer 1 - numerous fundamental papers on the topic are not considered, whose original findings are often repeated/confirmed in the papers mentioned here. The basic findings should therefore be explicitly mentioned in the text and then, if necessary, reference should be made to more recent reviews from 2011-2022 (in addition to the current reference #65), in which older important original papers on islet autoantibodies are cited and placed in context.

Regarding the current work of the T1DI consortium, which is indeed a significant source of data, there are some important publications from 2022 that should be addressed in the main article (e.g., PMID: 35803296, PMID: 35314671, PMID: 34758977; although some of these papers are currently cited in the supplement).

The condensed list of the main content-related statements/themes for group 1 (prior diagnosis) papers in Table 1 is good and helpful and should also be done in this form for the papers of groups 2-4. This could perhaps be supplemented in Table 1 (probably with a shorter list of core statements, which reflects the situation well). For reasons of balance between groups 1-4, the current Tables 2 and 3 should then perhaps also be moved to the supplementary material.

The checklist in Figure 3 is a very useful suggestion and it would be good if such a list became common in the future.

The references to table numbers in the text need to be updated.

Table 4: According to PMID: 31990315 (which is not considered in this review but probably should be), Fr1da is not a prospective birth cohort.

Reviewers' comments:

Reviewer #2 (Remarks to the Author):

As to the question “Is it possible to list the specific detection methods of islet autoantibodies in different studies and label the lab whether in above program or not in supplementary data?”, the author added the assays described and whether assays were noted to have been tested in a standardization program in Tables 2 and 3 (main text), and Supplemental Tables 1 and 2. As to the question “Is there any study identifying two profiles, such as autoantibody number in combination with its titer or affinity to predict the stage 3 T1D progression?”, the author answered almost all studies assessed an autoantibody feature, in addition to number, as summarized in Supplemental Table 1. In response to the reviewer comment, they have enhanced their discussion of this concept in the discussion in lines 460-462. As to the question “Is it possible to list the population in different studies in supplementary data?”, this has been added as Supplementary Table 3.

After revision, I believe it is a valuable systematic review which could provide important information for patients and care providers on what to expect for future type 1 diabetes progression. I recommend it to be accepted.

We appreciate the reviewer's close reading of and insight into this manuscript and our revision.

Reviewer #2 comments on response to reviewer #1):

I'm glad to provide help to review the manuscript and rebuttal letter again, and give some advice. I think the Reviewer 1's concerns have been suitably addressed by the authors, and I have some additional comments as follows:

1. Could the author label and note the 9 key articles identified by the group of experts that also met inclusion criteria in the method part?

In response to this suggestion, these papers are now denoted by an asterix in their respective tables and this designation is also now described in the text (lines 160-164).

2. As to the exclusion of LADA, I think author should also simply explain in method part, and should state in the discussion if the main results of this paper could also replicate in some of LADA studies instead of future study/review, apart from explaining the reason exclusion of LADA.

As recommended, an explanation for exclusion of unclearly classified populations, including LADA, is included in the methods (lines 155-158). We agree with the reviewer's point that it is possible that these findings may be replicated in adult populations as the subject of future studies (lines 607-608).

Reviewer #3 (Remarks to the Author):

This is a comprehensive and informative review article. The topic is timely and important.

Overall, the authors have adequately addressed the reviewers' comments.

The limited analysis of publications due to the time window 2011-2022 nevertheless remains a certain problem, since - as criticised by reviewer 1 - numerous fundamental papers on the topic are not considered, whose original findings are often repeated/confirmed in the papers mentioned here. The basic findings should therefore be explicitly mentioned in the text and then, if necessary, reference should be made to more recent reviews from 2011-2022 (in addition to the current reference #65), in which older important original papers on islet autoantibodies are cited and placed in context.

We thank Reviewer 3 for the review and additional suggestions for improvement. We agree that establishing a timeframe will inevitably exclude certain, seminal papers before and after the search limits, which in our opinion were unfortunately necessary given feasibility concerns (the current review already required review of over 11,000 abstracts). Furthermore, given changes in methods for autoantibody testing, we would also consider inclusion of more recent papers using newer methodology a strength. We thought that discussion of seminal works not found to meet inclusion criteria but contributing to our understanding of the topic was an excellent suggestion. To this end, we have included discussion of papers published since the completion of the search, including papers from the Fr1da and T1DI study groups (lines 589-602) and have further added context to our current work by referencing a comprehensive narrative review of islet autoantibody history, and commenting on how the findings of this systematic review support findings from earlier seminal papers (lines 581-587).

Regarding the current work of the T1DI consortium, which is indeed a significant source of data, there are some important publications from 2022 that should be addressed in the main article (e.g., PMID: 35803296, PMID: 35314671, PMID: 34758977; although some of these papers are currently cited in the supplement).

We agree that the T1DI consortium is a powerful cohort and its analyses will have significant implications. As suggested, this is now highlighted in the text and the Ghalwash et al paper referenced is now specifically mentioned (lines 589-602). Other listed T1DI papers fit our review criteria and were previously included as part of our systematic review; however, given the unique nature of this cohort and the power of harmonization, conclusions drawn from T1DI papers previously referenced in the supplement due to space are now described and referenced in the main text (lines 589-602).

The condensed list of the main content-related statements/themes for group 1 (prior diagnosis) papers in Table 1 is good and helpful and should also be done in this form for the papers of groups 2-4. This could perhaps be supplemented in Table 1 (probably with a shorter list of core statements, which reflects the situation well). For reasons of balance between groups 1-4, the current Tables 2 and 3 should then perhaps also be moved to the supplementary material.

Thank you for this feedback. Table 1 has now been expanded to include core statements summarizing all groups. Our ideal intent would be to include all tables in the main text. Unfortunately, this was impossible due to the large number of papers reviewed and journal size limitations. Although ideally all tables would be included, after discussion with the journal editor, we chose to prioritize accessibility of our content over balance between areas of review, and thus chose to keep tables 2 and 3 in the main text for better accessibility. Due to size, the journal has required us to keep supplementary tables 1 and 2 as supplementary data.

The checklist in Figure 3 is a very useful suggestion and it would be good if such a list became common in the future.

We agree and thank the reviewer for this feedback.

The references to table numbers in the text need to be updated.

Thank you for mentioning this oversight. References to tables have been updated in the main text.

Table 4: According to PMID: 31990315 (which is not considered in this review but probably should be), Fr1da is not a prospective birth cohort.

We thank the reviewer for identifying this error in categorization and we have revised the description of the Fr1da cohort. The paper mentioned, which we agree is of great importance to the field in general, was actually identified by our search; however, because the focus of this paper was prevalence of islet autoantibody positivity among the general population, rather than specific autoantibody features that help differentiate phenotypes of T1D, two reviewers voted to exclude the abstract from the analysis.

REVIEWERS' COMMENTS:

Reviewer #3 (Remarks to the Author):

The authors have responded adequately to the reviewers' concerns and improved their article. I have no further questions or comments.